# Score-based Membership Inference on Diffusion Models

## Abstract

Membership inference attacks (MIAs) against Diffusion Models (DMs) raise pressing privacy concerns by revealing whether a sample was part of the training set. While existing methods typically rely on measuring reconstruction error across multiple denoising steps as a test statistic, they often incur significant computational overhead. In this work, we present a simple yet successful attack statistic using only the predicted noise vectors from the DM's denoiser, or equivalently, the score. Specifically, we show that the expected denoiser output points toward a kernel-weighted local mean of nearby training samples, such that its norm encodes proximity to the training set and thereby reveals membership. Building on this observation, we propose SimA, a single-query attack that provides a principled, efficient alternative to existing multi-query methods. SimA consistently achieves ~~superior~~highly competitive performance across variants of DMs and the Latent Diffusion Models (LDMs) on eight different datasets. Its Monte Carlo variant (SimA-MC) exhibits state-of-the-art performance ~~across all~~in most experiments at 30 samples, significantly outperforming baseline methods in terms of TPR@1%FPR. These results demonstrate that complex reconstruction trajectories are unnecessary for effective membership inference, establishing SimA as a highly efficient benchmark for auditing privacy in DMs and LDMs.

## 1 Introduction

Generative image models leave evidence of their specific training data at deployment time in their generative process. While making draws from approximations of $p(x)$ or $p(x|y)$, they leave biases of the training samples (finite and fixed realizations from the real $p(x)$ or $p(x|y)$). These biases may be used in theory to reconstruct the training data, a process known as model inversion (Zhu et al., 2016; Creswell & Bharath, 2018; Carlini et al., 2023; Somepalli et al., 2023a;b; Gu et al., 2023).

The ability to invert these models raises concerns in privacy and intellectual property spaces for specific use-cases of generative models, but also possibly provides unique perspectives into the idiosyncrasies of the generative models themselves. If the models were perfect, they would sample from a distribution indistinguishable from the data generating process; the ways in which they deviate from this distribution inform upon their structure.

A critical precursor to model inversion is the *membership inference attack* (MIA), which determines whether a given image was included in the training set. MIA effectively constructs a classifier for identifying training examples, setting aside the problem of searching the domain for high-likelihood examples. While this may be the easier sub-problem of full model inversion, it is no less important.~~, as without successful MIA, model inversion is impossible.~~ MIA often serves as a foundational step or strong indicator of vulnerability for downstream data extraction.

Recent MIA statistics on diffusion models (Duan et al., 2023; Zhai et al., 2024; Kong et al., 2023; Matsumoto et al., 2023; Carlini et al., 2023) rely on either measuring reconstruction error along the Stochastic Differential Equation (SDE) trajectory or likelihood estimation (Fu et al., 2023). We argue that it incurs an unnecessary computational overhead. We demonstrate that using solely the output of DM's denoiser $\hat{\varepsilon}_\theta$ is sufficient to match or even surpass current state-of-the-art (SoTA) attack performance. Our method is based on a simple observation that the empirical distribution of an over-trained model will have peaks at each training

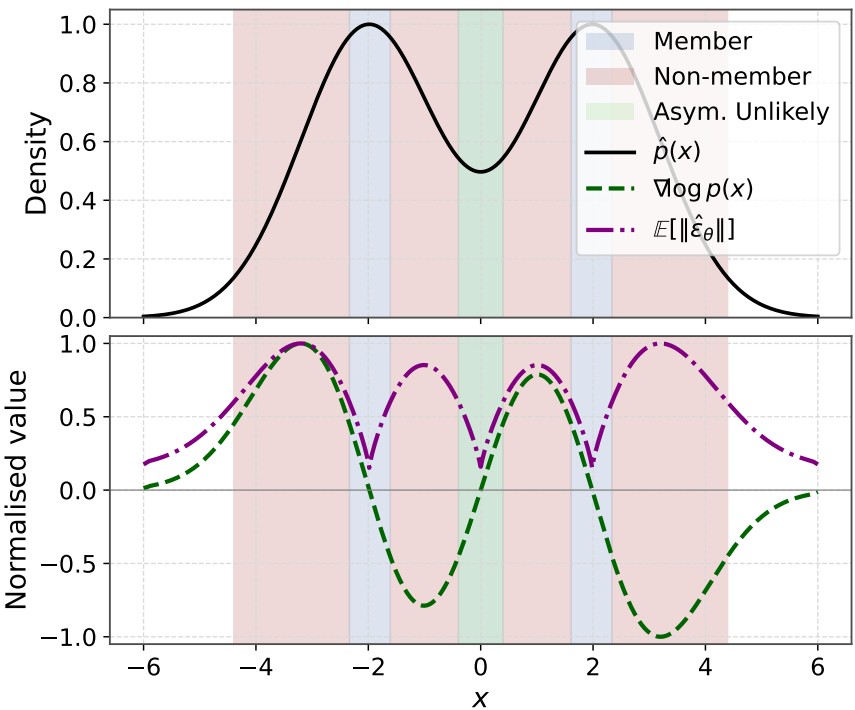

Figure 1: A diagram of our Membership Inference method in one dimension. In blue are regions of high membership likelihood, corresponding to low $\|\hat{\varepsilon}_\theta\|$, plotted in purple. The green region is unlikely to be sampled in high dimensions (c.f. Sec. 4).

point, and its gradient vanishes at the smoothed density (Figure 1), which exposes the membership signal. Unlike other generative models, this observation is particularly pertinent to diffusion-based models, where the model output $\hat{\epsilon}_\theta$ explicitly reveals the score function ($\hat{\epsilon}_\theta \approx -\sigma_t \nabla_{x_t} \log p_t(x_t)$) by design (Song et al., 2020; Ho et al., 2020). Hence, our attack statistic is simply the norm of the estimated score at the test point $x$ across diffusion times. More intuitions and a theoretical justification are described in Section 4.

Our empirical results show this simple method (**SimA**) has AUC scores ~~at~~around or above the previous SoTA for DDPM-weightsets (Ho et al., 2020) on common smaller datasets, and well above the previous SoTA for the related Guided Diffusion model (Dhariwal & Nichol, 2021) and vanilla LDM (Rombach et al., 2022) ~~on ImageNet-1k~~in terms of ASR and AUC. For large-scale LDMs such as Stable Diffusion, it achieves comparable results to the SoTA. To enhance stability, we further propose its Monte Carlo extension (**SimA-MC**). ~~While involving a higher computational cost due to additional model queries, SimA-MC further advances upon the current SoTA across all experiments, notably boosting the TPR@1%FPR.~~The marginal performance gains from additional queries in SimA-MC diminish as the dimensionality of the data increases. Fundamentally, Monte Carlo estimation in higher-dimensional spaces requires a proportionally larger number of samples. However, Latent Diffusion Models (LDMs) circumvent this issue by operating in a highly compressed latent space. SimA-MC in LDMs maintains its efficacy on high-resolution images without a prohibitive increase in query cost. Overall, SimA-MC with 30 MC samples achieves state-of-the-art performance in most experiments, particularly in terms of TPR@1%FPR. Meanwhile, SimA serves as an effective point-estimate surrogate that attains highly competitive performance with only a single query.

In summary, our contributions are the following:

1. A detailed derivation and theoretical justification of a simple score-based Membership Inference Attack method (SimA and SimA-MC), which is a reduction of other methods into a general framework.

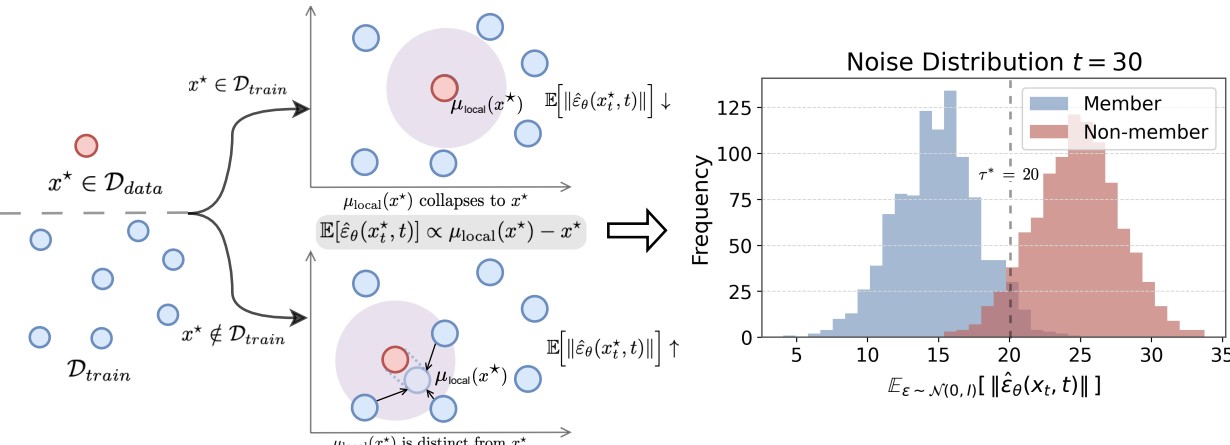

Figure 2: **Score-based MIA intuition with local-mean geometry.** In a small neighborhood ("local ball") around a query $x^\star$, let $\mu_{\text{local}}(x^\star)$ be the kernel-weighted mean of nearby training samples. The model's predicted noise (score) points from $x^\star$ toward this local mean, $\mathbb{E}[\hat{\varepsilon}_\theta(x_t^\star, t)] \propto \mu_{\text{local}}(x^\star) - x^\star$. For members $(x^\star \in \mathcal{D}_{train})$, the local mean $\mu_{\text{local}}(x^\star)$ collapses to training sample $x^\star$, producing small norms, whereas for non-members $(x^\star \notin \mathcal{D}_{\text{train}})$, $\mu_{\text{local}}(x^\star)$ deviates from $x^\star$, yielding larger norms. Right: the histogram at $t = 30$ shows the separation in $\mathbb{E}_{\varepsilon \sim \mathcal{N}(0,I)}[\|\hat{\varepsilon}_\theta(x_t, t)\|]$.

2. An empirical demonstration that SimA and SimA-MC provide top performance on standard datasets and independently trained base models.

All data splits, model checkpoints, training/fine-tuning scripts, and testing code are released on our GitHub repository (withheld for review).

## 2 Background and Related Work

**Diffusion Models:** Our membership inference work is specific to diffusion-based generative image models. Originally introduced as score-based generative models (without the explicit connection to the Diffusion model) in Song & Ermon (2019), a very large number of publications have explored variations of these models since that point (Song & Ermon, 2019; Ho et al., 2020; Song et al., 2020; Nichol & Dhariwal, 2021; Dhariwal & Nichol, 2021; Rombach et al., 2022).

While each contribution has its own particular training paradigm and architecture, our attack applies to the broad class of models that estimate a gradient flow field $\hat{\varepsilon}(x, t)$ at points $x$ for a smoothing parameter/diffusion time $t$ that approximates the gradient of the smoothed log-likelihood, $\nabla \log(p(x) * \mathcal{K}(t))$ (Kamb & Ganguli (2024), or Appendix A.4), which is often induced by a conceptual and/or training-time "forward" noise process $x_t = \sqrt{\bar{\alpha}_t} x_0 + \sigma_t \varepsilon$ and a "backward" de-noising process similar to denoising auto-encoder processes (Alain & Bengio, 2014). These may be variance-preserving or variance-exploding (Song et al., 2020), based on the exact parameterization of the noise schedule. In this work we directly use weights from the following models: DDPM (Ho et al., 2020), Guided Diffusion (Dhariwal & Nichol, 2021), Latent Diffusion Model (Esser et al., 2021), and Stable Diffusion (Rombach et al., 2022). Each of these directly estimate the noise vector $\varepsilon$ using a neural network $\hat{\varepsilon}(x, t)$. Our method manipulates $x$ and $t$ and analyze the networks' outputs, but otherwise is agnostic to the exact architecture and weights.

Latent Diffusion Models (Esser et al., 2021; Rombach et al., 2022) perform their forward process on the latent space of some encoder-decoder structure (usually a Variational Auto-encoder (Kingma et al., 2013)). Rao et al. (2025) has shown, this encoder-decoder structure appears to be closely related to membership inference.

The analytic tractability of these models permits theory about their behavior (Kamb & Ganguli, 2024; Lukoianov et al., 2025), even pre-dating the publication of the DDPM, e.g., Alain & Bengio (2014). Our

method is conceptually linked to results and intuition from Alain & Bengio (2014) and Kamb & Ganguli (2024).

**Membership Inference Attack threat models:** Model inversion and membership inference attacks predate the introduction of the DDPM and LDM generative image models; MIA was defined originally for general classification tasks (Shokri et al., 2017). A non-trivial amount of literature since that point has focused on MIA for generative image models (Chen et al., 2020) starting with GANs, due to its attack surface (pixel arrays) and its clear privacy and intellectual property implications.

Much of the terminology and structure were defined in the security context, where a threat model defines the scope and allowable resources for an attack vector. So-called "black box" attacks from the original MIA literature are performed without knowledge of the model weights or structure, and can only rely on input and output pairs from a deployed model. In contrast, "white box" attacks (Pang et al., 2023) have access to the full architecture/weight set. We consider the most common class of diffusion model MIAs, "grey-box" attacks (Duan et al., 2023; Zhai et al., 2024; Kong et al., 2023; Matsumoto et al., 2023; Carlini et al., 2023; Fu et al., 2023), which span a range of options between those two extrema; in general they have access to weights and/or internal representations, and may query the model for particular test points. We review each of these grey box attacks in detail in Section 4 and compare to each, with the exception of Zhai et al. (2024) as it performs membership inference on conditional generative models instead of the unconditioned case. Moreover, the present comparison is restricted to norm-based statistics for both SimA and the baselines. We chose to limit our analyses to this smaller and simpler class of statistics as classifier-based methods present a slightly different scenario, with classifier training/validation data and hyper-parameters additionally complicating analyses. SimA can be naturally extended to a classifier-based variant by training a classifier directly on the output vector rather than its norm.

**Membership Inference on Other Modalities:** MIA has also been investigated in other modalities, including language models as well as text-to-image and text-to-video generation (Shi et al., 2024; Li et al., 2024; 2026; Hu et al., 2025). Shi et al. (2024) compute the average log-likelihood of the $k\%$ of tokens with the lowest predicted probabilities, where a higher likelihood indicates a higher probability of membership. Li et al. (2024) and Li et al. (2026) instead infer membership from the entropy of the output token distribution. While these methods conform to the standard MIA paradigm, they rely on token-level likelihood signals that are specific to auto-regressive generation. In contrast, our method is tailored to diffusion models in the image domain and exploits their intrinsic training objective, namely score matching. Although we focus on unconditional image diffusion models, our method can in principle be extended to detecting training images of text-conditioned diffusion models.

## 3 Preliminaries

### Threat Model

We adopt the *grey-box* threat model standard in the diffusion-model MIA literature (Duan et al., 2023; Kong et al., 2023; Matsumoto et al., 2023; Fu et al., 2023), wherein an attacker attempts to discern between in-training and out-of-training data (all drawn from the same unknown data generating process). The attacker has the ability to query the denoiser $\hat{\varepsilon}_\theta(x, t)$ at arbitrary inputs $x$ and timesteps $t$ and observe the full output vector. The attacker does *not* receive access to model weights, gradients, internal activations, the training procedure, nor any extensive knowledge of $D_{\text{train}}$, but a general model class and input/output structure can be assumed; in particular, no shadow or reference models are trained. For LDMs, the attacker is additionally allowed access to the public encoder $\mathcal{E}$ to map the query image into the latent space; for text-conditioned models (e.g., Stable Diffusion), the attacker supplies the caption paired with the query image as the conditioning input. This access model is directly realistic for the rapidly growing class of *open-weight* models (e.g., Stable Diffusion), where the denoiser, encoder, and noise schedule are publicly released. For API-only deployments that expose neither intermediate denoiser outputs nor timestep control, our attack–

like all grey-box diffusion MIAs–is not directly applicable. We view our results primarily as a privacy-auditing tool for model developers and as a lower bound on the leakage of publicly released checkpoints.

**Diffusion models and notation:** We consider diffusion-based generative models trained with the denoising objective of Ho et al. (2020). Let $\{\beta_t\}_{t=1}^T$ denote the variance schedule of the forward process, and define $\alpha_t := 1 - \beta_t$, $\bar{\alpha}_t := \prod_{s=1}^t \alpha_s$, and $\sigma_t^2 := 1 - \bar{\alpha}_t$. Under the variance-preserving (VP) forward process, a clean sample $x_0 \sim p_{\text{data}}$ is perturbed as $x_t = \sqrt{\bar{\alpha}_t}\, x_0 + \sigma_t \varepsilon$ with $\varepsilon \sim \mathcal{N}(0, I)$. For Latent Diffusion Models (LDMs) (Rombach et al., 2022), the same process is defined on the latent space of a pre-trained encoder–decoder pair $(\mathcal{E}, \mathcal{D})$, and all quantities above are understood to apply to $z_0 = \mathcal{E}(x_0)$.

# 4 Methodology

## Method Overview

The predicted noise $\hat{\varepsilon}_\theta$ is outputted by the neural network, which is a scaled estimator of the score $-\sigma_t \nabla_{x_t} \log p_t(x_t)$ (Song et al., 2020; Ho et al., 2020; Luo, 2022), for data generating distribution $p(x)$ and its mollifications $p_t(x)$ on a noising schedule $\sigma_t$. Our Simple Attack (**SimA**) method is

$$\mathcal{A}(x, t) = \|\hat{\varepsilon}_\theta(x, t)\|. \tag{1}$$

Here, $x$ is the test point, which ostensibly was drawn from $p(x)$, but is either in the training data or not, and $t$ is the diffusion time parameter. We provide a more rigorous derivation and its connections to other MIA methods in the following sections, but we feel that its intuition is also instructive on why such a simple method would work.

Figure 2 visualizes the intuition of our method: the expected norm of the predicted score for a query image $x^\star$ at time $t$ is effectively the gradient of a Gaussian kernel density estimator (see Appendix D). For well-separated points, these estimators' implicit distributions will have peaks at each training point; the gradient vanishes at critical points of the smoothed density, which include the peaks corresponding to the original data points (the blue region in the Fig. 1), up to a bias term from the smoothing. Manipulating this fact allows us to form a simple yet successful estimator.

There should be false positive terms at the other critical points (the green region in Fig 1). In high dimension, these should be by-in-large saddle points between maxima. Because these points occur directly between original datapoints at their arithmetic mean, if the data manifold (the support of $p(x)$) has any curvature, these points would be off manifold; empirically we do not seem to encounter many of them, as indicated by the TPR@1%FPR measurements (see Section 5).

## 4.1 From Forward Diffusion to Score

~~**Notation:** Let $\{\beta_t\}_{t=1}^T$ be the variance schedule of the forward diffusion process in DDPM (Ho et al., 2020). Define $\alpha_t := 1 - \beta_t$ and its cumulative product $\bar{\alpha}_t := \prod_{s=1}^t \alpha_s$. The total noise variance accumulated up to step $t$ is $\sigma_t^2 := 1 - \bar{\alpha}_t$.~~

**Forward Diffusion as a Scaled Gaussian Convolution:** With the variance-preserving (VP) schedule of Ho et al. (2020), the forward model is

$$x_t = \sqrt{\bar{\alpha}_t}\, x_0 + \sigma_t \varepsilon, \quad \varepsilon \sim \mathcal{N}(0, I),\ \sigma_t^2 = 1 - \bar{\alpha}_t, \tag{2}$$

where $x_0 \sim p_{\text{data}}$. Marginalizing $x_0$ gives the perturbed data distribution (Song & Ermon, 2019). For a *clean* (without noise) sample $x \in \mathbb{R}^d$,

$$p_t(x) = \int_{\mathbb{R}^d} p_{\text{data}}(x_0)\, \mathcal{N}\big(x \mid \sqrt{\bar{\alpha}_t} x_0, \sigma_t^2 I\big)\, dx_0. \tag{3}$$

Equation 3 is a Gaussian convolution of the original data distribution (see Appendix D for an explicit derivation), followed by a global scaling by $\sqrt{\bar{\alpha}_t}$.

$$p_t(x) \;=\; \frac{1}{\bar{\alpha}_t^{d/2}} \left( p_{\text{data}} * \mathcal{N}\!\left(0, \tfrac{\sigma_t^2}{\bar{\alpha}_t} I\right)\right)\!\left(\frac{x}{\sqrt{\bar{\alpha}_t}}\right). \tag{4}$$

Therefore, $p_t(x)$ is the data distribution convolved with scaled Gaussian distribution whose kernel's covariance $\frac{\sigma_t^2}{\bar{\alpha}_t} I = (\bar{\alpha}_t^{-1} - 1)I$ grows monotonically with the timestep $t$.

**Gradient of $\mathbf{p_t(x)}$:** Writing the kernel in standard form $\mathcal{K}_t(x, x_0) = (2\pi\sigma_t^2)^{-d/2}\exp(-\|x - \sqrt{\bar{\alpha}_t}x_0\|^2/2\sigma_t^2)$, we can compute its spatial gradient: $\nabla_x \mathcal{K}_t = -\sigma_t^{-2}(x - \sqrt{\bar{\alpha}_t}x_0)\mathcal{K}_t$. We then combine this with Eq. 3 to obtain

$$\nabla_x p_t(x) = -\sigma_t^{-2} \int p_{\text{data}}(x_0)(x - \sqrt{\bar{\alpha}_t}x_0)\mathcal{K}_t(x, x_0)\,dx_0 \tag{5}$$

which requires continuity assumptions on $p_{\text{data}}$ which are usually assumed by DDPM analyses (Alain & Bengio, 2014).

**Introducing exact likelihood of each datapoint** $q_t(x_0 \mid x)$**:** Define the exact distribution of the $x_0$ given an observation from the Gaussian smoothed distribution:

$$q_t(x_0 \mid x) = \frac{p_{\text{data}}(x_0)\,\mathcal{K}_t(x, x_0)}{p_t(x)}. \tag{6}$$

Because $p_t(x)$ normalizes Eq. 3, we can then rewrite Eq. 5 as

$$\nabla_x p_t(x) = -\frac{p_t(x)}{\sigma_t^2}\left[ x - \sqrt{\bar{\alpha}_t}\underbrace{\mathbb{E}_{q_t(x_0|x)}[x_0|x]}_{\mu_t(x)} \right]. \tag{7}$$

We call $\mathbb{E}_{q_t(x_0|x)}[x_0|x] = \mu_t(x)$ the *denoising mean*; it is the likelihood-weighted average of the positions of the datapoints that could have generated $x$ at time $t$ through the forward process, which is the same as the mean optimal solution to the denoising problem.

**Obtaining the exact score:** Dividing Eq. equation 7 by $p_t(x)$ yields the score of distribution $p_t(x)$ at $x$ (Lemma 6 of Pidstrigach (2022), or, alternatively, applying the chain rule to $\nabla_x \log p_t(x)$ and then substituting in values):

$$s_t(x) = \nabla_x \log p_t(x) = -\frac{x - \sqrt{\bar{\alpha}_t}\,\mu_t(x)}{\sigma_t^2}. \tag{8}$$

This score function $s_t(x)$ is the desired output of the $\varepsilon$-parameterization of Score-based Denoising (Song et al., 2020) and the original DDPM (Ho et al., 2020). During training the UNet is asked to predict the standard noise (Ho et al., 2020):

$$\hat{\varepsilon}_\theta(x, t) \;\approx\; \frac{x_t - \sqrt{\bar{\alpha}_t}\,\mu_t(x)}{\sigma_t} \;=\; -\sigma_t \nabla_x \log p_t(x) \tag{9}$$

This is consistent with Eq. 151 of Luo (2022). For $x \in \text{supp}\, p_t$ (with or without noise), the estimator $\hat{\varepsilon}_\theta(x, t)$ should approximate the negative score of $p_t(x)$. However, in practice the data distribution $p_{\text{data}}$ becomes the empirical distribution $p_{\text{training}}$, which is a finite sample of points. The noised distribution $p_t(x)$ is then a kernel smoothing of that empirical distribution, and its finite-sample denoising mean is described in Kamb & Ganguli (2024) and refined in Lukoianov et al. (2025). Equation 3 of Lukoianov et al. (2025) states it as:

$$\mu_t^{\text{finite}}(x) = \sum_{i=1}^{N} w_i(x, t)x_0^{(i)}, \quad w_i(x, t) = \text{softmax}_i\left\{ -\frac{1}{2\sigma_t^2}\|x - \sqrt{\bar{\alpha}}x_0^{(j)}\|_2^2 \right\}_{j\in[N]} \tag{10}$$

where the $x_0^{(i)}$ are training data, and $\text{softmax}_i$ is the $i^{\text{th}}$ index of a softmax function over the $N$ training data points. The discrepancy between the finite sample optimal denoised $x$ distribution and the large sample limite $q_t(x_0|x)$ gives rise to our membership inference attack; the model "overfits" to the training set, and that overfitting gap is the discrepancy which **SimA** seeks to exploit.

## 4.2 Membership Inference Attack

Given a datapoint $x \in \mathbb{R}^d$ and a $t = 1, \ldots, T$, our membership decision criterion $\mathcal{A}$ is defined as

$$\mathcal{A}(x, t) = \|\hat{\varepsilon}_\theta(x, t)\|_p. \tag{11}$$

Using $\ell_p$ norms other than $p = 2$ provide slightly improved performance. This trend is also found in another MIA method, Kong et al. (2023). While this is somewhat mysterious, the $\ell_4$ norm appears in sum-of-squares computations (Barak et al., 2015), spherical harmonics (Stanton & Weinstein, 1981), and blind source separation (Hyvarinen, 1997) with surprising regularity.

Following the Bayes-optimal loss–threshold formulation of membership inference in classification models by Sablayrolles et al. (2019), we recast the decision rule for diffusion models. Specifically, we define

$$\mathcal{M}_{\mathrm{opt}}(x, t) = \mathbb{1}\big[\mathcal{A}(x, t) \leq \tau\big], \tag{12}$$

where $\tau$ is a threshold calibrated on a held-out validation set and $\mathcal{A}(x, t) = \|\hat{\varepsilon}_\theta(x, t)\|_2$ is the estimated noise at $x$ for diffusion step $t$. If the predicted noise norm is *smaller* than $\tau$, the sample is inferred to be a *member*; otherwise, it is classified as a *non-member*.

**Monte Carlo variant of SimA:** Notably, SimA is a scaled point estimate ($\varepsilon = 0$) of our intuition in Figure 2. Although it is analytically derived in Section 4.1, point estimate of this nature is inherently numerically unstable. Given that both the score function and its estimator are smooth, we can make it more stable and overall better by local averaging. This leads to the MC solution. Formally, given a test image $x$ and $N$ Monte Carlo (MC) samples $\varepsilon_n \sim \mathcal{N}(0, I)$, the Monte Carlo Variant of SimA (**SimA-MC**) is defined as

$$\mathcal{A}_{\mathrm{mc}}(x, t) = \frac{1}{N} \sum_n \|\hat{\varepsilon}_\theta(x_t, t)\| \quad x_t = \sqrt{\bar{\alpha}_t}\, x_0 + \sigma_t \varepsilon_n. \tag{13}$$

We originally thought that such a high-dimensional Monte Carlo would perform poorly until a very large number of samples. However, this method turns out to be surprisingly tractable. The SimA-MC method provides SoTA performance within 30 samples, often within 10. For sufficient ($n \in [10, 30]$) samples, it improves all performance metrics on all datasets of our experiments. The results is summarized in lower block of all tables in Section 5.

## 4.3 Theoretical Justification on Three Cases

The attack criterion $\mathcal{A}$ can be applied to three cases, which we expand upon below (see appendix E for detailed derivation of the first two cases).

**Case 1 — Member of the Training Set:** Let $x^{(k)}$ denote one of the training images $\{x^{(i)}\}_{i=1}^N$. As $t \to 0$, the finite sample denoising mean collapses to the input (full derivation is in case 1 of Appendix E):

$$\mu_t^{\mathrm{finite}}(x^{(k)}) \xrightarrow[t \to 0]{} x^{(k)} \tag{14}$$

Consequently, the estimated noise vector shrinks to zero as well, meaning our criterion $\mathcal{A}(x^{(k)}, t \to 0)$ should be small:

$$\boxed{\hat{\varepsilon}_\theta(x^{(k)}, t) = \frac{x^{(k)} - \sqrt{\bar{\alpha}_t}\, x^{(k)}}{\sigma_t} \sim \frac{\sigma_t}{2} x^{(k)} \xrightarrow[t \to 0]{} 0.} \tag{15}$$

While the actual value at zero is undefined, and values for small $t < 10$ are empirically unstable for non-member $x^{(k)}$ (leading to a poor estimator, see Appendix F for the detailed reason), we find that for $t \in [10, 300]$, these values will still be smaller than the case 2. These time frames are unfortunately noise schedule and data dependent.

**Case 2 — Held-out but On-Manifold:** Here $x^\dagger$ is sampled from the same data distribution $p_{\text{data}}$ as the training set, yet was *never* shown during training. We can consider only the local means (i.e. $\mu_t^{\text{finite}}(x^\dagger) \approx \mu_t^{\text{local}}(x^\dagger)$) because we already established that the diffusion model acts as a KDE (Eq. 10 in Section 4.1). This means that many points will be "far" from our chosen point, and thus have exponentially decreasing influence. The only high influence points will be "close" points, which is definitionally the local neighborhood.

We use the notation of local moment matching (Bengio et al., 2012; Alain & Bengio, 2014) to describe these points. Consider the local mean defined with respect to that $B_r(x^\dagger)$:

$$\mu_t^{\text{local}}(x^\dagger) = m_r(x^\dagger) = \int_{B_r(x^\dagger)} x p_t(x|x^\dagger) dx \tag{16}$$

where $r \asymp \sigma_t/\sqrt{\bar\alpha_t}$. Theorems 2 and 3 of Alain & Bengio (2014) put together produce a statement about a term that is equivalent to $\hat\varepsilon$ (Eq. 28 of Alain & Bengio (2014)) (Full derivation is in case 2 of Appendix E):

$$\boxed{m_r(x^\dagger) - x^\dagger \approx \frac{r^2}{d+2} \left.\frac{\partial \log p_t(x)}{\partial x}\right|_{x^\dagger} = \frac{r^2}{d+2}\left(\frac{x^\dagger - \sqrt{\bar\alpha_t}\mu_t(x^\dagger)}{\sigma_t^2}\right) = -\frac{r^2}{\sigma_t(d+2)}\hat\varepsilon_\theta(x^\dagger, t)} \tag{17}$$

We claim that for in-support regions of Gaussian mollified empirical distributions with well separated points, these regions will generally not be flat, and thus $\|\hat\varepsilon_\theta(x^\dagger, t)\|$ will tend away from zero. The magnitude to which $\|\hat\varepsilon_\theta(x^\dagger, t)\|$ diverges from zero clearly depends on the maximal density of the dataset, but as high dimensional spaces have exponentially larger volumes than lower dimensional spaces, even for large datasets (e.g. ImageNet in ResNet resolution) we can expect these voids to be non-trivially large.

**Case 3 — Out-of-Distribution (OOD).** Since no training data support is available in out-of-distribution (OOD) regions, the diffusion model lacks information about these areas. Consequently, the learned score field in such regions is necessarily an extrapolation, and the theoretical derivations established under the in-distribution assumption no longer hold. We do not expect either the diffusion model or our attack criteion to perform well in these regions. First, even though the theoretical finite-sample optimal denoiser is well defined (Eq. 10), a neural network approximation to it will have very little training data in these regions. Second, a trial datapoint $x^*$ is by-definition not from $p_{data}$ in these regions.

### 4.4 SimA in Comparison to Other Diffusion MIA Models

A standard concept in MIA is the use of the training loss function evaluated on the data points in question as the member/non-member decision criterion. This is the Loss criterion, and is proposed in Matsumoto et al. (2023). They use a stochastic sample $\varepsilon$ to estimate this criterion, adding it to the test point $x^*$.

$$\text{Loss} = \|\varepsilon - \hat\varepsilon_\theta(\sqrt{\bar\alpha_t}x^* + \sqrt{1-\bar\alpha_t}\varepsilon, t)\| \quad \text{(Matsumoto et al., 2023)} \tag{18}$$

In theory this should be evaluated across a large number of $\varepsilon$ measurements, but for each point they choose to use only one. This method is predicated on the idea that for points in the training set, the noise estimation will be better or even overfit in comparison to points not in the training set. SimA is mathematically the scaled evaluation of this method at $\varepsilon = 0$, which is the mean and mode of the $\varepsilon$ distribution, but inherently very different and performs much better. Effectively, Matsumoto et al. (2023) are measuring draws from lower likelihood areas, which may not exhibit the overfit phenomenon as well as $\varepsilon = 0$. Fu et al. (2023) provides this same loss estimate as the selection criterion, but increases the Monte Carlo sampling to 20 and performs it only for a single step, sampling $\varepsilon_t$ the stepwise noise instead of sampling $\varepsilon$.

SecMI of Duan et al. (2023) takes this one step further, evaluating not only a score term but also a single step term, which measures sensitivity to single step differences in $t$:

$$\text{SecMI} = \|\sqrt{1-\bar\alpha}(\hat\varepsilon_\theta(x^*, t) - \hat\varepsilon_\theta(\sqrt{\bar\alpha_{t+1}}x^* + \sqrt{1-\bar\alpha_{t+1}}\varepsilon, t+1))\| \tag{19}$$

SecMI is dependent on sampling $\varepsilon$'s as well; the authors prescribe using $N = 12$ samples.

The closely related PIA method computes this same loss quantity again, but using a $t = 0$ term instead of the $\varepsilon$ sample.

$$\text{PIA} = \|\hat{\varepsilon}_\theta(x^*, t = 0) - \hat{\varepsilon}_\theta(\sqrt{\bar{\alpha}_t}x^* + \sqrt{1 - \bar{\alpha}_t}\hat{\varepsilon}_\theta(x^*, t = 0), t)\| \quad (20)$$

While in theory diffusion models might not be well defined at $t = 0$, in practice they often can extrapolate as they are trained on nearby $t$; in the continuous time case they are trained on the interval $[0, 1]$. Our method has similar components to this method, manipulating $t$ around the test point, but again replacing its "ground-truth" noise with $\varepsilon = 0$ (here replaced by $\hat{\varepsilon}_\theta(x^*, t = 0)$).

**Remarkably, unlike the three aforementioned attack statistics that rely on reconstruction along the SDE trajectory, SimA uniquely leverages the behavior of the score function based the observation in Figure 1.**

## 5    Experiments

### 5.1    Environment

All experiments were conducted on a dedicated computing node running the Ubuntu 22.04.4 LTS operating system. The hardware infrastructure is equipped with an AMD EPYC 7453 28-Core Processor, paired with 256 GB of system RAM. Hardware acceleration for model training and evaluation was provided by ~~eight~~nine NVIDIA A40 GPUs. The software environment was built using Python 3.11.3, with all deep learning workflows and tensor computations implemented via PyTorch version 2.10.0, utilizing CUDA toolkit 12.8. ~~More software library details are provided on our online resource in the Appendix ??.~~

### 5.2    Setup

We evaluated our attack on ~~15 member–held-out pairs~~ drawn from ~~11 datasets~~nine datasets with different DMs and LDMs (see Appendix B). The experiments were conducted on the following target models:

**Denoising Diffusion Probabilistic Model:** For CIFAR-10, CIFAR-100 (Krizhevsky et al., 2009), STL10-U (unlabeled split) (Coates et al., 2011), and CelebA (Liu et al., 2015), we trained a vanilla DDPM (Ho et al., 2020) from scratch on the *member* set. From each training split we subsample $n$ images and partition them equally into a *member* set and a *held-out* set.

**Latent Diffusion Model:** For CIFAR-10, CelebA (Liu et al., 2015) and ImageNet-1K (Russakovsky et al., 2015), we trained a vanilla LDM (Rombach et al., 2022) from scratch on the *member* set. Likewise, we subsample $n$ images from each training split and partition them equally into a *member* set and a *held-out* set.

**Pre-trained Guided Diffusion:** We examined the publicly released Guided Diffusion model[1] (Dhariwal & Nichol, 2021) trained on ImageNet-1K (Russakovsky et al., 2015). Since there is no confirmed information that this checkpoint was previously trained on ImageNet-1K validation set, ImageNetV2 (Recht et al., 2019), collected to mirror the original distribution (same data collection process and same year range), serves as the *held-out* set.

**Pre-trained Latent Diffusion Models:** For Pokémon[2], COCO2017-Val Lin et al. (2014), and Flickr30k Young et al. (2014), we fine-tune Stable Diffusion v1-4[3] on a randomly selected subset of each training split, reserving an equally sized subset as the **held-out** set. We also studied the original Stable Diffusion v1-5[4] checkpoint, pre-trained on LAION-Aesthetics v2 5+ (Schuhmann et al., 2022) (*member* set). Here we sampled 2500 images from LAION-2B-MultiTranslated[5] as non-members, respectively. Notably, the images from LAION-2B-MultiTranslated are filtered with attributes *pwatermark* $< 0.5$; *prediction (aesthetic_score)* $> 5.0$

---

[1]`https://github.com/openai/guided-diffusion`

[2]`https://huggingface.co/datasets/lambdalabs/pokemon-blip-captions`

[3]`https://huggingface.co/CompVis/stable-diffusion-v1-4`

[4]`https://huggingface.co/stable-diffusion-v1-5/stable-diffusion-v1-5`

[5]`https://huggingface.co/datasets/laion/laion2B-multi-joined-translated-to-en`

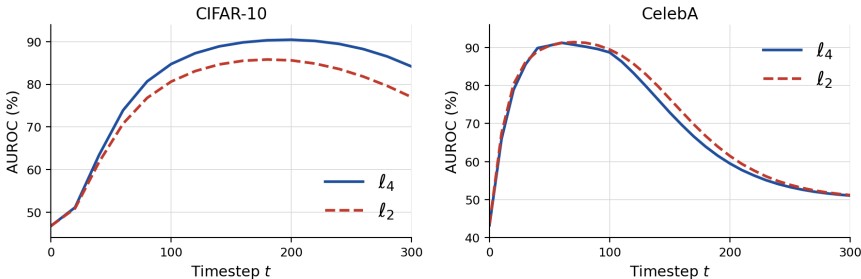

Figure 3: AUROC (%) comparison of $\ell_4$ and $\ell_2$ norms across timesteps on CIFAR-10 and CelebA.

and *similarity* > 0.3. *pwatermark* and *prediction* are to minimize the domain shift between the member set and the held-out set. And *similarity* is to ensure the alignment of the text-image pairs.

Full statistics including dataset splits, resolutions, conditions, etc. are summarized in Table 5 of Appendix B.

**Baselines:** Earlier membership-inference attacks (MIAs) aimed at GANs and VAEs—e.g., (Chen et al., 2020; Hilprecht et al., 2019; Hu & Pang, 2021)—do not transfer well to diffusion models, as shown by Duan et al. (2023). Consequently, we restricted our evaluation to attacks specifically designed for diffusion models. We compared our method with four baselines: SecMI (Duan et al., 2023), PIA (Kong et al., 2023), PFAMI (Fu et al., 2023), Loss (Matsumoto et al., 2023). We omitted CLiD (Zhai et al., 2024), a text-conditioned MIA, because its text-supervision is incompatible with the setup used in most of our experiments, and the provided code is not usable.

~~Evaulation~~Evaluation **Metrics:** We evaluated attack performance using several metrics: ASR (attack success rate, i.e., membership inference accuracy), AUC (Area Under ROC Curve), TPR at 1% FPR (TPR@1%FPR), and the number of queries per attack (#Query). We further provide a comprehensive computational comparison of different baseline methods in terms of GFlops and Latency in Appendix G. The TPR@1%FPR is computed by selecting the threshold $\tau$ at which the false positive rate falls just below 0.01, and reporting the corresponding true positive rate at that operating point. The ASR is defined as the maximum accuracy achieved over all thresholds, i.e. $\tau^* = \max_\tau \frac{1}{2}\big(\mathrm{TPR}(\tau) + 1 - \mathrm{FPR}(\tau)\big)$ in the balanced setting. The AUC is computed as the trapezoidal integral of $\mathrm{TPR}(\tau)$ against $\mathrm{FPR}(\tau)$ across all thresholds.

**Implementation details:** Some baselines (Duan et al., 2023; Kong et al., 2023; Zhai et al., 2024) augmented their score- or feature-vector statistic with an auxiliary neural classifier; to focus on the statistic itself we evaluated only the norm-based versions, which were SecMI$_\mathrm{stat}$ and PFAMI$_\mathrm{Met}$. Of the two variants (PIA and PIAN) introduced by Kong et al. (2023), we benchmarked only PIA, as PIAN showed no statistically significant gain in general from their experiments.

To minimise re-implementation error, our codebase reused the official releases of SecMI, PIA, and Guided Diffusion wherever possible. We failed to re-use PFAMI's code as the provided code was inoperable. A reimplemented copy is provided in our code base. PFAMI, in several cases, failed to attack the victim model (ASR $\approx 50\%$). We hypothesize that this degradation arose from the sensitivity of its Monte Carlo estimator to the effective sample size, which varied with dataset characteristics and latent dimensionality, yielding high-variance estimates. For each method, we followed the hyperparameter suggestion in their original paper. Notably, $l_2$-norm, $l_4$-norm and $l_2$-norm were used for SecMI, PIA and Loss as suggested. $l_4$-norm was used for SimA as it achieved the best performance in general. Figure 3 shows an ablation on norms choice.~~For the timestep-dependent attacks (SimA, SecMI, PIA, Loss), we swept $t = 0{:}300$ and reported the best-performing value for each method (PFAMI is timestep-free and therefore needed no sweep).~~ More implementation details can be found in Section 5.3.

**Calibration on $t$ and $\tau$:** The member and non-member sets were each randomly partitioned into two disjoint subsets of equal size, one used for validation and the other for testing. For the timestep-dependent attacks (SimA, SecMI, PIA, Loss), we swept $t = 0{:}300$ on the validation subset and chose the best-performing

Table 1: Performance of benchmark methods on **DDPM** across four datasets. In the upper block, the **bold** values indicate the best results (baselines and SimA). In the lower block, the **bold** values indicate the new SoTA achieved by SimA-MC compared to the whole upper block.

| Method | #Query↓ | CIFAR-10 (%) | | | CIFAR-100 (%) | | | STL10-U (%) | | | CelebA (%) | | |
|---|---|---|---|---|---|---|---|---|---|---|---|---|---|
| | | ASR↑ | AUC↑ | TPR$^\dagger$↑ | ASR↑ | AUC↑ | TPR$^\dagger$↑ | ASR↑ | AUC↑ | TPR$^\dagger$↑ | ASR↑ | AUC↑ | TPR$^\dagger$↑ |
| PIA | 2 | **85.23** | **92.03** | 29.27 | 83.14 | **90.42** | 29.68 | 79.04 | 86.49 | 23.28 | 83.54 | 91.40 | 41.27 |
| PFAMI$_{met}$ | 20 | 73.87 | 80.20 | 8.00 | 70.52 | 77.40 | 8.27 | 64.6 | 70.99 | 5.70 | 71.71 | 79.58 | 16.32 |
| SecMI$_{stat}$ | 12 | 82.40 | 89.84 | 33.16 | 81.50 | 88.5 | 33.47 | 77.85 | 85.24 | 21.71 | 83.73 | 91.46 | 31.77 |
| Loss | 1 | 77.66 | 84.47 | 23.83 | 75.42 | 82.95 | 20.65 | 73.14 | 79.31 | 17.44 | 79.05 | 87.33 | 32.59 |
| SimA (⋆) | 1 | 83.69 | 90.23 | **35.67** | **82.92** | 90.15 | **37.52** | **79.27** | **86.68** | **29.1** | 83.79 | 91.75 | 42.40 |
| SimA-MC (#mc=10) | 10 | 84.32 | 91.29 | 43.49 | 83.03 | 89.9 | 43.39 | 79.31 | 86.90 | 34.12 | 87.1 | 94.06 | 59.01 |
| SimA-MC (#mc=30) | 30 | 85.14 | 92.44 | 44.64 | 83.67 | 91.16 | 44.40 | 80.63 | 88.03 | 37.44 | 87.88 | 94.99 | 60.19 |

$^\dagger$ True Positive Rate at $1\%$ False Positive Rate.

Table 2: Attack performance of four baseline methods used to evaluate memorization of **LDMs**. In the upper block, the **bold** values indicate the best results (baselines and SimA). In the lower block, the **bold** values indicate the new SoTA achieved by SimA-MC compared to the whole upper block.

| Method | #Query↓ | CIFAR-10 (%) | | | CelebA (%) | | | ImageNet-1K (%) | | |
|---|---|---|---|---|---|---|---|---|---|---|
| | | AUC↑ | ASR↑ | TPR@1%FPR↑ | AUC↑ | ASR↑ | TPR@1%FPR↑ | AUC↑ | ASR↑ | TPR@1%FPR↑ |
| PIA | 2 | 85.76 | 78.50 | 15.70 | 83.37 | 76.01 | 9.48 | 65.88 | 62.21 | 2.99 |
| PFAMI$_{mat}$ | 20 | 74.77 | 69.93 | 7.84 | 75.66 | 68.10 | 7.91 | 67.34 | 62.94 | 3.89 |
| SecMI$_{stat}$ | 12 | 87.53 | 79.97 | 18.86 | 83.33 | 75.94 | 10.53 | 68.50 | 63.40 | 3.70 |
| Loss | 1 | 73.48 | 67.31 | 7.60 | 68.56 | 63.47 | 6.16 | 67.23 | 63.09 | **4.01** |
| SimA (⋆) | 1 | **89.34** | **81.58** | **19.75** | **84.87** | **77.14** | **10.92** | **69.58** | **64.91** | 3.75 |
| SimA-MC (#mc=10) | 10 | 90.47 | 82.94 | 39.71 | 91.65 | 83.31 | 39.64 | 71.15 | 66.13 | 7.38 |
| SimA-MC (#mc=30) | 30 | 91.12 | 83.83 | 41.74 | 93.12 | 84.40 | 42.01 | 72.07 | 67.34 | 7.90 |

$t$ for each method (PFAMI is timestep-free and therefore needed no sweep). Then we applied this $t$ on the testing subset. We also calibrate $\tau$ on the validation subset and applied it to the testing set.

### 5.3 Main results

**In Table 1**, all DDPM baselines were trained on a member split and evaluated on an held-out split of equal sizes: $25\,\text{k}/25\,\text{k}$ for CIFAR-10 and CIFAR-100, $50\,\text{k}/50\,\text{k}$ for STL10-U, and $30\,\text{k}/30\,\text{k}$ for CelebA, all at a spatial resolution of $32 \times 32$. We adopted the public checkpoints and splits released by SecMI for CIFAR-10/100, and retrained STL10-U and CelebA for $18\,\text{k}$ and $60\,\text{k}$ steps, respectively. Across all datasets and metrics, SimA achieved the best in almost all the experiments over three metrics (ASR, AUC, TPR@1%FPR) while requiring the fewest queries, underscoring its efficiency and practical advantage. ~~Sim-MC~~ SimA-MC reaches the performance of SimA within 10 MC samples and outperforms within 30 MC samples. The performance of SimA-MC is expected to asymptotically approach that of SimA and eventually outperform

Table 3: Performance of public checkpoint of **Guided Diffusion** pretrained on ImageNet1K. In the upper block, the **bold** values indicate the best results (baselines and SimA). *Member* set: ImageNet-1K (train split); *Held-out*: ImageNetV2 (train split).

| Method | #Query↓ | ImageNet | | |
|---|---|---|---|---|
| | | ASR↑ | AUC↑ | TPR@1%FPR↑ |
| PIA | 2 | 64.61 | 66.45 | 9.89 |
| PFAMI$_{Met}$ | 20 | 67.54 | 72.35 | 4.03 |
| SecMI$_{stat}$ | 12 | 78.02 | 82.70 | **34.48** |
| Loss | 1 | 57.57 | 60.68 | 6.92 |
| SimA (⋆) | 1 | **85.84** | **89.56** | 21.55 |
| SimA-MC (#mc=10) | 10 | 81.10 | 85.79 | 10.62 |
| SimA-MC (#mc=30) | 30 | 83.46 | 87.01 | 17.53 |

Table 4: Attack performance of four baseline methods used to evaluate memorization of **Stable Diffusion**. In the upper block, the **bold** values indicate the best results (baselines and SimA). In the lower block, the **bold** values indicate the new SoTA achieved by SimA-MC compared to the whole upper block.

| Method | #Query↓ | Pokémon (%) | | | MS-COCO (%) | | | Flickr30K (%) | | |
|---|---|---|---|---|---|---|---|---|---|---|
| | | AUC↑ | ASR↑ | TPR@1%FPR↑ | AUC↑ | ASR↑ | TPR@1%FPR↑ | AUC↑ | ASR↑ | TPR@1%FPR↑ |
| PIA | 2 | **94.02** | **89.98** | 31.67 | 92.57 | 85.86 | 24.80 | 68.57 | 64.35 | 2.74 |
| PFAMI$_{met}$ | 20 | 50.0 | 23.14 | 0.0 | 85.82 | 78.48 | 12.08 | 64.91 | 61.11 | 2.47 |
| SecMI$_{stat}$ | 12 | 87.89 | 80.78 | 29.85 | 91.50 | 83.85 | **38.93** | **72.02** | **66.29** | **5.12** |
| Loss | **1** | 92.33 | 85.77 | **41.34** | 83.02 | 75.97 | 13.39 | 63.03 | 60.00 | 2.46 |
| SimA ($\star$) | **1** | 94.42 | 87.91 | 20.59 | **94.02** | **87.17** | 30.09 | 70.01 | 65.70 | 2.71 |
| SimA-MC (#mc=10) | 10 | **96.84** | **91.43** | **63.49** | 93.33 | 85.79 | **45.42** | 70.54 | 65.83 | 4.41 |
| SimA-MC (#mc=30) | 30 | **96.83** | **92.01** | **70.33** | **94.17** | **86.78** | **52.41** | **72.43** | **66.97** | **6.21** |

it as the number of Monte Carlo samples increases. Notably, the Monte Carlo variant ~~boost~~boosts the performance at the critical region, as suggested by TPR@1%FPR.

**In Table 2**, the setting is mostly identical to Table 1. We increased the spatial resolution of CelebA to $64 \times 64$. For ImageNet-1K, the LDM was trained on 100k images of its train split. We take another 100k images as the *held-out* set. SimA achieves the best performance with a single query, and ~~Sim-MC~~SimA-MC advances the SoTA within 10 Monte Carlo Samples.

**In Table 3**, we further evaluated Guided Diffusion (Dhariwal & Nichol, 2021) at their public class-conditional ImageNet-1K checkpoints ($256 \times 256$). SimA has the highest ASR and AUC by a wide margin, while SecMI has a very high TPR@1%FPR. SimA-MC exhibits suboptimal performance in this experiment ~~, likely due to the vast sampling space ($3 \times 256 \times 256$) in the image domain. Effectively covering such a high-dimensional manifold requires a substantially larger number of Monte Carlo samples, which poses a significant challenge in terms of computational tractability. However, Sim-MC works pretty well in a latent space diffusion model as shown in our other experiments~~. In most cases, mc_samples = 10 is sufficient to match or exceed the performance of SimA. However, this behavior is strongly dependent on the data dimensionality. Very high-dimensional settings, such as Guided Diffusion ($3 \times 256 \times 256$), require a substantially larger number of MC samples, which is computationally prohibitive in our setting. This curse of dimensionality is alleviated when moving to Latent Diffusion Models (LDMs), where the effective dimensionality is significantly reduced; this is precisely why SimA-MC remains effective on Stable Diffusion, a variant of LDMs.

**Table 4** follows the experimental protocol of Zhai et al. (2024) and evaluates membership-inference attacks on Stable Diffusion under two scenarios:

*Fine-tuning.* A Stable Diffusion v1-4 checkpoint is fine-tuned on the designated *member* split of each target dataset; attacks are launched on paired *member/held-out* splits.

*Pre-training.* A pre-trained Stable Diffusion v1-5 model is attacked directly, without additional fine-tuning. The *member* set is a subset of LAION-Aesthetics v2 5+ that was used during pre-training, while the *held-out* set is an equally-sized split drawn from the target dataset.

For every dataset we create five random *member/held-out* partitions. For LAION-2B-MultiTranslated, we provided the results for both unconditional and text-conditional cases. All other experiments are run in unconditional modes. The unconditional baseline is obtained by passing an empty string to the CLIP text encoder. The main paper reports results for Pokémon(fine-tuning), MS-COCO (fine-tuning), Flickr30k (fine-tuning); LAION-2B-MultiTranslated (pre-training) appear in Table 6 of the Appendix C.

This series of experiments indicates that Diffusion processes on their own can already be solved for MIA to a high degree of fidelity with a very simple estimator.

## 6 Conclusion

In the present work we have described a simple membership inference estimator, giving theoretical justification for its performance, and for the performance of similar estimators in the literature which previously

lacked a unified theoretical backing. We demonstrate that this estimator has competitive performance on many baselines. Overall, we believe SimA can serve as a lightweight auditing tool to evaluate the model memorization and privacy leakage.

## 7 Broader Impacts and Ethics

While it serves as a lightweight privacy auditing tool for model developers to quantify memorization, our developments could theoretically be deployed by malicious actors. Our work serves to warn open-weight model developers about possible risks, and may lead to responsible disclosure practices for developers. The immediate risk of our own work is outweighed by the risk of future model releases without public knowledge of inversion possibilities. In particular, since we do not provide a complete inversion system (we do not consider the search problem), this particular work has limited risk, but does make the open-weight community aware of the possibility of development for such a system being developed.

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

## A    Appendix

## B    Datasets and splits

Datasets and splits used for our experiments are summarized in Table 5.

## C    Stable Diffusion pre-trained on LAION-Aesthetics v2 5+.

We also studied the original Stable Diffusion v1-5[6] checkpoint, pre-trained on LAION-Aesthetics v2 5+ (Schuhmann et al., 2022) (*member* set). Here we sampled 2500 images from LAION-2B-MultiTranslated[7] as non-members, respectively. Notably, the images from LAION-2B-MultiTranslated are filtered with attributes *pwatermark < 0.5; prediction (aesthetic_score) > 5.0* and *similarity > 0.3. pwatermark* and *prediction* are to minimize the domain shift between the member set and the held-out set. And *similarity* is to ensure the alignment of the text-image pairs. We provided both unconditional and text-conditional cases in Table 6. PIA performs the best at this setting, and SimA achieved comparable results.

## D    From marginal density to explicit Gaussian convolution

Let $\bar{\alpha}_t = \prod_{s=1}^t \alpha_s$ and $\sigma_t^2 = 1 - \bar{\alpha}_t$. The VP forward marginal is

$$p_t(x) = \int_{\mathbb{R}^d} p_{\text{data}}(x_0)\, \mathcal{N}\big(x \,\big|\, \sqrt{\bar{\alpha}_t}\, x_0,\ \sigma_t^2 I\big)\, dx_0. \tag{21}$$

Set $u = \sqrt{\bar{\alpha}_t}\, x_0$ so that $x_0 = u/\sqrt{\bar{\alpha}_t}$ and $dx_0 = \bar{\alpha}_t^{-d/2}\, du$. Then

$$p_t(x) = \bar{\alpha}_t^{-d/2} \int_{\mathbb{R}^d} p_{\text{data}}\left(\frac{u}{\sqrt{\bar{\alpha}_t}}\right)\, \mathcal{N}\big(x \,\big|\, u,\ \sigma_t^2 I\big)\, du. \tag{22}$$

Introduce $\tilde{x} := x/\sqrt{\bar{\alpha}_t}$ and $\tilde{u} := u/\sqrt{\bar{\alpha}_t}$, so $u = \sqrt{\bar{\alpha}_t}\, \tilde{u}$ and $du = \bar{\alpha}_t^{d/2}\, d\tilde{u}$. Substituting into equation 22 cancels the Jacobians and yields

$$p_t(x) = \int_{\mathbb{R}^d} p_{\text{data}}(\tilde{u})\, \mathcal{N}\big(\sqrt{\bar{\alpha}_t}\, (\tilde{x} - \tilde{u}) \,\big|\, 0,\ \sigma_t^2 I\big)\, d\tilde{u}. \tag{23}$$

Use the Gaussian scaling identity

$$\mathcal{N}(az \mid 0,\ \sigma^2 I) = a^{-d} \mathcal{N}\left(z \,\middle|\, 0,\ \frac{\sigma^2}{a^2} I\right) \qquad \text{(for } a > 0\text{)},$$

with $a = \sqrt{\bar{\alpha}_t}$ and $z = \tilde{x} - \tilde{u}$. Then equation 23 becomes

$$p_t(x) = \bar{\alpha}_t^{-d/2} \int_{\mathbb{R}^d} p_{\text{data}}(\tilde{u})\, \mathcal{N}\left(\tilde{x} - \tilde{u} \,\middle|\, 0,\ \frac{\sigma_t^2}{\bar{\alpha}_t} I\right)\, d\tilde{u}. \tag{24}$$

The integral in equation 24 is a (Euclidean) convolution evaluated at $\tilde{x}$:

$$\big(p_{\text{data}} * \mathcal{N}(0, \tfrac{\sigma_t^2}{\bar{\alpha}_t} I)\big)(\tilde{x}) = \int_{\mathbb{R}^d} p_{\text{data}}(\tilde{u})\, \mathcal{N}\left(\tilde{x} - \tilde{u} \,\middle|\, 0,\ \tfrac{\sigma_t^2}{\bar{\alpha}_t} I\right)\, d\tilde{u}.$$

Therefore,

$$p_t(x) = \bar{\alpha}_t^{-d/2} \left(p_{\text{data}} * \mathcal{N}\big(0, \tfrac{\sigma_t^2}{\bar{\alpha}_t} I\big)\right)\left(\frac{x}{\sqrt{\bar{\alpha}_t}}\right) \tag{25}$$

with $\sigma_t^2/\bar{\alpha}_t = \bar{\alpha}_t^{-1} - 1$.

---

[6] https://huggingface.co/stable-diffusion-v1-5/stable-diffusion-v1-5
[7] https://huggingface.co/datasets/laion/laion2B-multi-joined-translated-to-en

Table 5: Datasets and splits used for our experiments.

| Model | Member | Held-out | Pre-trained | Fine-tuned | Splits | Resolution | Cond. |
|---|---|---|---|---|---|---|---|
| DDPM | CIFAR-10 | CIFAR-10 | No | – | 25k/25k | 32 | – |
| | CIFAR-100 | CIFAR-100 | No | – | 25k/25k | 32 | – |
| | STL10-U | STL10-U | No | – | 50k/50k | 32 | – |
| | CelebA | CelebA | No | – | 30k/30k | 32 | – |
| Latent Diffusion Model | CIFAR-10 | CIFAR-10 | No | – | 25k/25k | 32 | – |
| | CIFAR-100 | CIFAR-100 | No | – | 25k/25k | 32 | – |
| | STL10-U | STL10-U | No | – | 50k/50k | 32 | – |
| | CelebA | CelebA | No | – | 30k/30k | 64 | – |
| | ImageNet-1k | ImageNet-1k | No | – | 100k/100k | 256 | class |
| | ImageNet-1k | ImageNetv2 | No | – | 3k/3k | 256 | class |
| Guided Diffusion | ImageNet-1k | ImageNetV2 | Yes | No | 3k/3k | 256 | class |
| Stable Diffusion V1-4 | Pokémon | Pokémon | Yes | Yes | 416/417 | 512 | text |
| | COCO2017-Val | COCO2017-Val | Yes | Yes | 2.5k/2.5k | 512 | text |
| | Flickr30k | Flickr30k | Yes | Yes | 10k/10k | 512 | text |
| Stable Diffusion V1-5 | LAION-Aesthetics v2 5+ | LAION-2B-MultiTranslated | Yes | No | 2.5k/2.5k | 512 | text |
| | LAION-Aesthetics v2 5+ | COCO2017-Val | Yes | No | 2.5k/2.5k | 512 | text |

Table 6: Performance of **Stable Diffusion** that is ~~pretrianed~~pretrained on LAION-Aesthetics v2 5+. LAION-2B-MultiTranslated is the pre-training setting.

| Method | #Query↓ | LAION-2B-MultiTranslated (%) | | |
|---|---|---|---|---|
| | | ASR↑ | AUC↑ | TPR@1%FPR↑ |
| PIA | 2 | **61.1** | **63.8** | **4.24** |
| PFAMI$_\text{Met}$ | 20 | 50.0 | 40.8 | 1.11 |
| SecMI$_\text{stat}$ | 12 | 56.7 | 58.2 | 2.69 |
| Loss | **1** | 49.5 | 37.4 | 1.05 |
| SimA | **1** | 60.1 | 63.7 | 3.15 |

# E   MIA in three cases

When the *data prior* is the empirical distribution constructed from the training set $\{x^{(i)}\}_{i=1}^N$. We have $p_\text{train}(x_0) = \frac{1}{N}\sum_i \delta(x_0 - x^{(i)})$. For any test image $x$, the UNet predicts

$$\hat{\varepsilon}_\theta(x,t) \approx \frac{x - \sqrt{\bar{\alpha}_t}\,\mu_t(x)}{\sigma_t}, \tag{26}$$

We are interested in the quantity

$$\mu_t^\text{finite}(x) = \mathbb{E}_{q_t(x_0|x)}[x_0] \;=\; \sum_{i=1}^N w_i(x,t)\,x^{(i)},$$

$$w_i(x,t) = \frac{\exp\left[-\|x - \sqrt{\bar{\alpha}_t}\,x^{(i)}\|^2/(2\sigma_t^2)\right]}{\displaystyle\sum_{j=1}^N \exp\left[-\|x - \sqrt{\bar{\alpha}_t}\,x^{(j)}\|^2/(2\sigma_t^2)\right]}. \tag{27}$$

Equations equation 27 express the posterior mean as a weighted average of the training samples, where each kernel weight $w_i(x,t)$ depends on the Euclidean distance between the noisy query $x$ and the down-scaled datum $\sqrt{\bar{\alpha}_t}\,x^{(i)}$.

### E.1   Case 1 — Training member

Pick $x = x^{(k)} \in \{x^{(i)}\}$. Set $x = x^{(k)}$ in equation 27. Define the squared distances

$$d_{ik}(t) \; := \; \left\| x^{(k)} - \sqrt{\bar{\alpha}_t}\, x^{(i)} \right\|^2, \qquad \Delta_{ik}(t) \; := \; \frac{d_{ik}(t) - d_{kk}(t)}{2\sigma_t^2}. \tag{28}$$

Using equation 27 and equation 28 we obtain

$$w_k(x^{(k)}, t) = \left[ 1 + \sum_{i \neq k} \exp\!\left(-\Delta_{ik}(t)\right) \right]^{-1}, \tag{29}$$

$$w_{i \neq k}(x^{(k)}, t) = \exp\!\left(-\Delta_{ik}(t)\right) w_k(x^{(k)}, t). \tag{30}$$

**Small-noise limit** $t \to 0$.   Because $\sigma_t^2 = 1 - \bar{\alpha}_t \to 0$ and

$$1 - \sqrt{\bar{\alpha}_t} = O(\sigma_t^2), \qquad d_{kk}(t) = \left(1 - \sqrt{\bar{\alpha}_t}\right)^2 \|x^{(k)}\|^2 = O(\sigma_t^4),$$

we have, for $i \neq k$,

$$\Delta_{ik}(t) \;\sim\; \frac{\|x^{(k)} - x^{(i)}\|^2}{2\sigma_t^2} \xrightarrow[t \to 0]{} +\infty, \qquad \Delta_{kk}(t) = 0.$$

Hence

$$w_k(x^{(k)}, t) \xrightarrow[t \to 0]{} 1, \qquad w_{i \neq k}(x^{(k)}, t) \xrightarrow[t \to 0]{} 0. \tag{31}$$

This implies

$$\boxed{\mu_t^{\text{finite}}(x^{(k)}) \xrightarrow[t \to 0]{} x^{(k)}}. \tag{32}$$

Moreover, substituting into the estimator equation 26,

$$\left\| \hat{\varepsilon}_\theta(x^{(k)}, t) \right\|_2 \;\approx\; \left\| \frac{x^{(k)} - \sqrt{\bar{\alpha}_t}\, x^{(k)}}{\sigma_t} \right\|_2 \;=\; \frac{|1 - \sqrt{\bar{\alpha}_t}|}{\sigma_t} \|x^{(k)}\|_2. \tag{33}$$

To see the asymptotic form using Taylor expansion around $\sigma_t^2 = 0$, note that

$$\sqrt{\bar{\alpha}_t} = \sqrt{1 - \sigma_t^2} = 1 - \tfrac{1}{2}\sigma_t^2 - \tfrac{1}{8}\sigma_t^4 + O(\sigma_t^6),$$

so that

$$1 - \sqrt{\bar{\alpha}_t} = \tfrac{1}{2}\sigma_t^2 + O(\sigma_t^4).$$

Therefore

$$\frac{1 - \sqrt{\bar{\alpha}_t}}{\sigma_t} = \tfrac{1}{2}\sigma_t + O(\sigma_t^3),$$

and hence

$$\left\| \hat{\varepsilon}_\theta(x^{(k)}, t) \right\|_2 \;\sim\; \tfrac{\sigma_t}{2} \|x^{(k)}\|_2 \xrightarrow[t \to 0]{} 0. \tag{34}$$

### E.2   Case 2 — Held-out but On-Manifold

Consider a test data point $x^\dagger$ which is not in our original dataset (i.e., $x^\dagger \notin \{x^{(i)}\}_{i=1}^N$), but sampled from the same generating distribution $p_{\text{data}}$. Under the diffusion process assumptions, the local weighted mean $\mu_t(x)$ in equation 27 has the same algebraic form as a kernel regression (Nadaraya, 1964; Watson, 1964), using training dataset $\{x^{(i)}\}_{i=1}^N$ with (effective) Gaussian bandwidth

$$h(t) \; := \; \frac{\sigma_t}{\sqrt{\bar{\alpha}_t}}. \tag{35}$$

The weights $w_i(x, t)$ in equation 27 are proportional to $\exp\left(-\|x^\dagger - \sqrt{\bar\alpha} x^{(i)}\|_2^2 / (2h(t)^2)\right)$, so $\mu_t^{\text{finite}}(x^\dagger)$ coincides with a Gaussian–kernel local average of "nearby" training points, where "nearby" is on the order of bandwidth $h(t)$. The kernel-weighted local mean with radius $r$ around $x^\dagger$ is defined as

$$m_r(x^\dagger) := \frac{\displaystyle\int_{B_r(x^\dagger)} u \, K_r(u - x^\dagger) \, p(u) \, du}{\displaystyle\int_{B_r(x^\dagger)} K_r(u - x^\dagger) \, p(u) \, du} \approx \mu_t^{\text{finite}}(x^\dagger), \tag{36}$$

$$K_r(z) \propto \exp\left(-\frac{\|z\|^2}{2r^2}\right), \tag{37}$$

where $r \asymp h(t)$ and $p(u)$ is the empirical distribution. This is a normalization of the local mean defined in Eq. 16 of this paper, and Appendix 6.4.1 of Alain & Bengio (2014). By local moment matching (Bengio et al., 2012; Alain & Bengio, 2014), combining Theorems 2 and 3 of Alain & Bengio (2014) (see their Eq. (28)) yields the second-order expansion

$$m_r(x^\dagger) - x^\dagger = \frac{r^2}{d+2} \left.\frac{\partial \log p_t(x)}{\partial x}\right|_{x^\dagger} + o(r^3), \tag{38}$$

where $p_t$ is the Gaussian-mollified density at scale $\sigma_t$. Using equation 9, we obtain the proportionality stated in the main text:

$$\boxed{m_r(x^\dagger) - x^\dagger \approx \frac{r^2}{d+2} \left.\frac{\partial \log p_t(x)}{\partial x}\right|_{x^\dagger} = \frac{r^2}{d+2}\left(\frac{x^\dagger - \sqrt{\bar\alpha_t}\mu_t(x^\dagger)}{\sigma_t^2}\right) = -\frac{r^2}{\sigma_t(d+2)}\hat\varepsilon_\theta(x^\dagger, t)} \tag{39}$$

Notably, this equation should *not* be correct if $x^\dagger$ is in a low density region of the support of $p_{data}$, but these points should generally be rarely sampled by definition. Moreover, the $\hat\varepsilon$ estimate of the score will also be extrapolating at those points.

**Implication for the attack statistic.** In in-support regions of the empirical distribution, the local geometry is generally *not* flat, so $\|m_r(x^\dagger) - x^\dagger\| > 0$ for $r > 0$ sufficiently small. By equation 39, this implies $\|\hat\varepsilon_\theta(x^\dagger, t)\|$ is bounded away from zero (at fixed $t$), hence $\mathcal{A}(x^\dagger, t) = \|\hat\varepsilon_\theta(x^\dagger, t)\|_p$ exceeds the member case (Case 1). This matches the intuition summarized in the main text: on-manifold, held-out queries denoise less precisely than memorized training points, yielding a moderately larger attack statistic without the divergence seen off-manifold.

### E.3 Case 3 — Out-of-distribution (far off-manifold)

Since no training data support is available in out-of-distribution (OOD) regions, the diffusion model lacks information about these areas. Consequently, the learned score field in such regions is necessarily an extrapolation, and the theoretical derivations established under the in-distribution assumption no longer hold.

~~Specifically, in the adversarial setup, optimization proceeds in two steps per iteration. First, we update the VAE with the usual reconstruction+KL objective and, after a warm-up, add a non-saturating adversarial term (hinge form; equivalent to $-\mathbb{E}[D(\hat x)]$) gated to turn on only after a chosen global-step threshold (we set this to 2,000 steps in our runs). The adversarial term is weighted adaptively using a VQGAN-style~~ (Esser et al., 2021) ~~gradient-norm ratio on the decoder's last layer and then scaled by a constant factor of 0.8; before the start step, its weight is zero by design. Second, we update the discriminator with a hinge loss on real images $x$ versus reconstructions $\hat x$ recomputed in a no-grad branch (to avoid generator gradient leakage), using AdamW with the same learning rate/betas/weight-decay as the VAE optimizer.~~

## F   Why Extremely Early and Late $t$ are Problematic:

Intuitively, $\hat\varepsilon_\theta(x, t)|_{t=0}$ is expected to achieve the best performance. Because the noise added to a member at $t = 0$ is expected to be zero (as shown in **case 1**) and non-zero for a non-member. However, in region of low

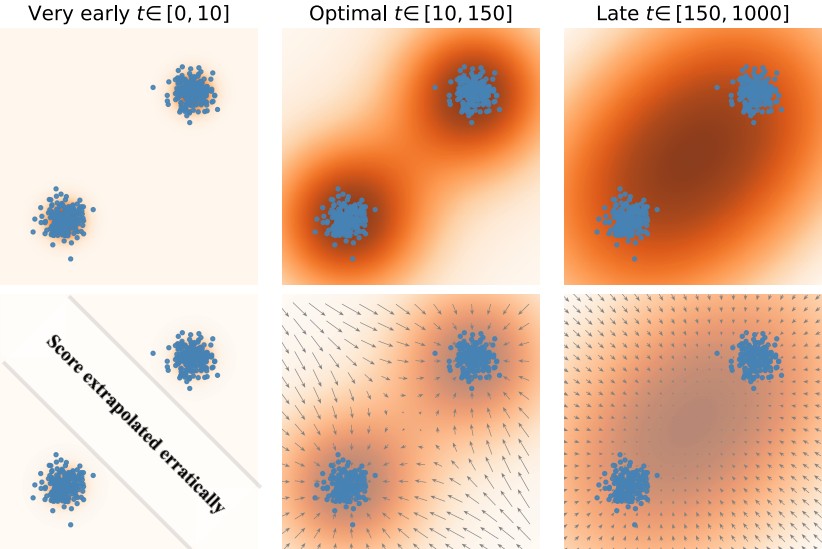

Figure 4: Top: density $p_t(x)$; blue dots are training samples. Bottom: estimated score $\nabla_x \log p_t(x)$. For **very early** $t \in [0, 10]$, the inter-mode region is low-density, so the score extrapolates erratically (shaded band). **Optimal** $t \in [10, 300]$**:** moderate Gaussian convolution enlarges the support and regularizes the estimator—density bridges the modes and the score points coherently toward them, yielding the strongest separation between members and held-out points. For **late** $t \in [300, 1000]$, $p_t$ approaches an isotropic Gaussian and the score collapses toward the global mean, diminishing membership signal.

data density, score-matching lacks sufficient evidence to *reliably* estimate the score function (Song & Ermon, 2019). Song & Ermon (2019) argues that training minimizes the expected value of score estimates (here is $\mathbb{E}_{p_{t=0}}\big[\|\hat{\varepsilon}_\theta(x_{t=0}, t) - \varepsilon\|^2\big]$), which provides *inaccurate scores* where $p_{t=0}(x)$ is infinitesimal. To be specific, for the input $\big\{x \in \mathcal{R} \mid \{\mathbf{x}_i\}_{i=1}^N \cap \mathcal{R} = \varnothing, \quad \mathcal{R} \subset \mathbb{R}^d\big\}$, $\nabla_x p_{t=0}(x)$ extrapolates erratically. Consequently, for **very early timesteps** ($\sigma_t \ll 1$) the learned field outside the tightly supported member set can plateau or even shrink, nullifying the privacy signal. Increasing $t$ corresponds to *extra Gaussian convolution*, expanding the effective support and regularising the score. Figure 5 plots the average normalised estimator magnitude $\|\hat{\varepsilon}_\theta(x, t)\|$ for $t \in [0, 300]$ on the *member* and *held-out* splits across datasets. Transient fluctuations are confined to the very earliest timesteps ($t \approx 0$), and the maximal gap between the curves typically occurs at early—but not initial—timesteps, indicating that moderately early diffusion steps provide the strongest membership signal.

Conversely, for **late steps** ($\sigma_t \approx 1$) the forward process approaches an isotropic Gaussian (data information gradually diminish); $p_t$ is nearly homogeneous, so the posterior $\mu_t(x) = q_t(x_0 \sim p_{\text{training}} \mid x)$ collapses to $\mu_t(x) = q_t(x_0 \sim \mathcal{N}(0, I) \mid x)$, which depends on test images and membership information is lost. Figure 4 illustrates the phenomenon. The *optimal* timestep $t^*$ is therefore dataset-specific and also depends on the noise schedule.

## G    Comparison of Query Budget, GFlops and Latency

Table 7 provides a comparison among baselines in terms of #queries, GFlops, and Latency on CIFAR-10 dataset.

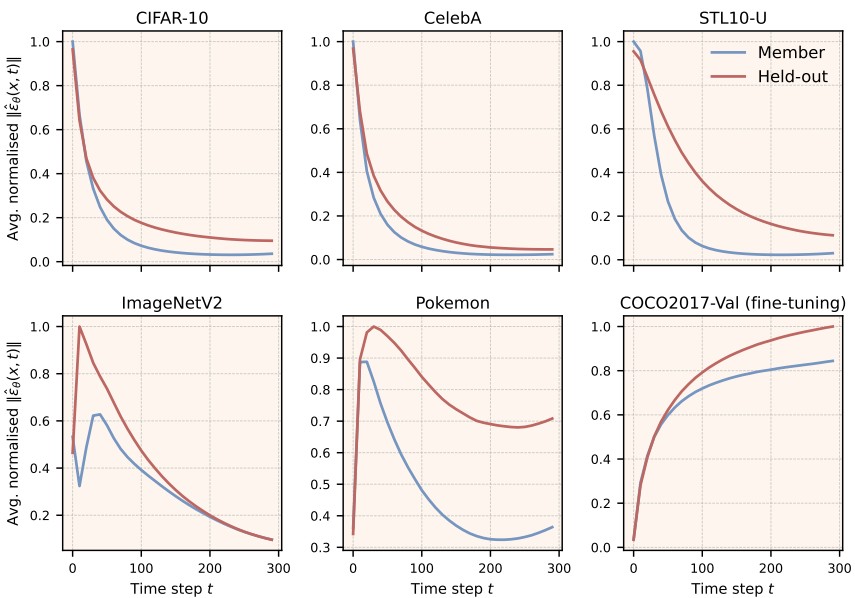

Figure 5: The average normalised estimator magnitude $||\hat{\varepsilon}_\theta(x, t)||$ for $t \in [0, 300]$ on the *member* and *held-out* splits across datasets

Table 7: Computational efficiency of MIA methods on CIFAR-10. UNet NFE denotes the number of UNet function evaluations per query image. GFLOPs are computed as UNet FLOPs per forward pass (measured via `fvcore`) multiplied by NFE. Although there might be other FLOPs besides the forward pass, the UNet call dominates the FLOPs. Wall-clock time is measured on a single GPU averaged over 50,000 samples.

| Method | UNet NFE ↓ | GFLOPs ↓ | Latency (ms/sample) ↓ |
|--------|-----------|----------|----------------------|
| SimA | 1 | 6.24 | 0.71 |
| Loss | 1 | 6.24 | 0.71 |
| PIA | 2 | 12.48 | 1.41 |
| SecMI | 12 | 74.87 | 8.45 |
| PFAMI | 20 | 124.78 | 14.08 |

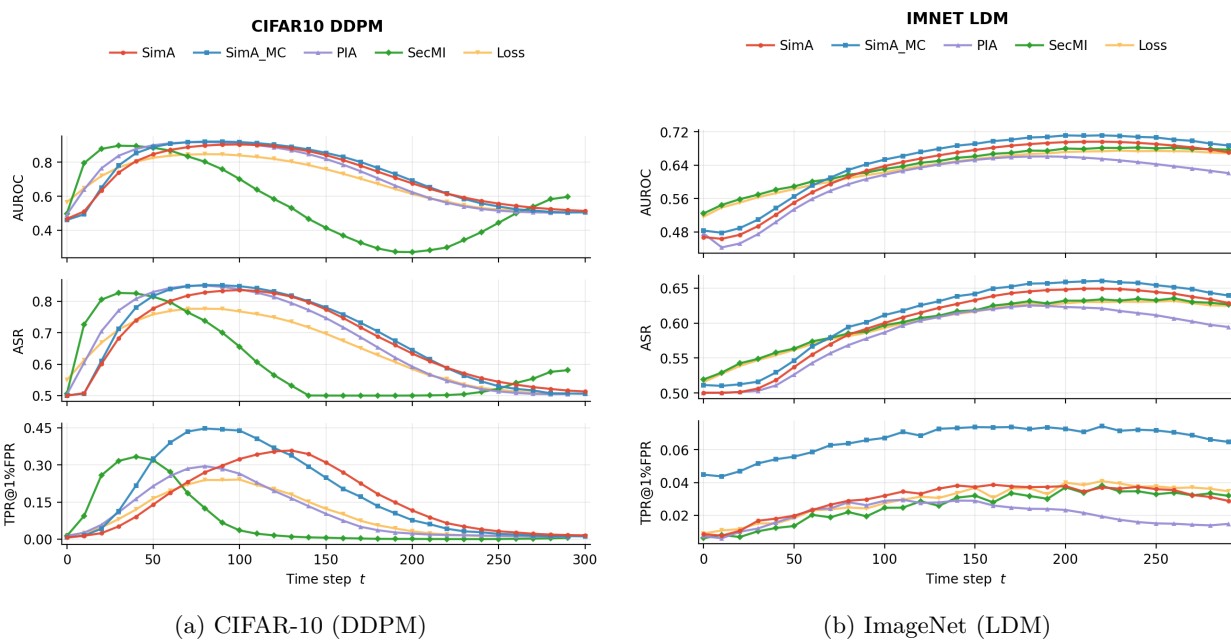

(a) CIFAR-10 (DDPM)        (b) ImageNet (LDM)

Figure 6: AUROC, ASR, and TPR@1%FPR of different membership inference attacks as a function of the diffusion time step $t$, on CIFAR-10 (DDPM) and ImageNet (LDM).

