# OpenReview forum: "Score-based Membership Inference on Diffusion Models"
_TMLR — Decision pending for TMLR_

### Review · Reviewer_AMdZ · 2026-05-10

**Summary Of Contributions:**

This paper proposes a score-based membership inference attack against diffusion models. The main observation is that the predicted noise vector of a diffusion model can be interpreted as a scaled score estimate, and that its norm can reveal whether a query sample lies close to a training point. Based on this observation, the authors introduce SimA, a single-query membership inference statistic based on the norm of the denoiser output, and SimA-MC, a Monte Carlo-smoothed variant that averages the statistic over noisy perturbations.

The main strength of the paper is the simplicity of the proposed statistic. The theoretical intuition is also appealing, especially the link between score estimation, kernel smoothing, and local training-set proximity. The experiments are reasonably broad in terms of model families and datasets, and the single-query nature of SimA is practically attractive.

The main weakness is that the current evidence does not fully support some of the strongest claims, since the advantage of the proposed method is not always significant. The experimental protocol also appears to sweep timesteps and report the best-performing value, which raises concerns about test-set tuning unless the selection is performed on a truly separate validation set. The theoretical argument is insightful but still relies on idealized assumptions about optimal denoisers, empirical KDE-like behavior, local separation, and in-distribution queries; these assumptions should be made more explicit and better connected to the actual trained neural networks.

The overall recommendation is **Accept with minor revision**.

**Audience:**

Yes

**Audience Explanation:**

Yes. The paper should be of interest to researchers working on diffusion models, privacy auditing, membership inference, memorization, and generative-model security.

**Broader Impact Concerns:**

This work provides a lightweight privacy auditing tool for diffusion models and could help model developers detect memorization or training-data leakage.
However, the same method can be used to infer whether a particular image, face, or sensitive private record was included in a model’s training set.

**Claims And Evidence:**

Yes

**Claims Explanation:**

Yes, but partially.
The evidence supports the weaker and more important claim that the denoiser-output norm is a strong and efficient MIA statistic for diffusion models. The reported results show that SimA is competitive with prior methods while using one query, and SimA-MC often improves TPR@1%FPR substantially. This is a meaningful empirical finding.
However, the paper should temper claims such as “consistently achieves superior performance” or “state-of-the-art across all experiments.” SimA is often strong but usually not the best; SimA-MC improves many results but can be worse in the Guided Diffusion setting; and in the LAION pretraining setting, the appendix states that PIA performs best while SimA is comparable. These results are still valuable, but the wording should reflect the actual evidence.

**Requested Changes:**

1. **Clarify the threat model.**
The paper should explicitly state what access the attacker needs: denoiser outputs, arbitrary timestep queries, model weights, latent representations, or conditioning information. The practical relevance for open-weight models versus API-only models should also be discussed.
2. **Add a preliminary / problem formulation section.**
I recommend adding a short preliminary section before the main analysis. This section should formally define the research object, threat model, membership inference goal, diffusion-model notation, attack statistic, query budget, and evaluation protocol. Currently, the paper introduces many notations and assumptions before they are fully defined, which may make the paper accessible only to readers already highly familiar with diffusion-model MIAs.
3. **Fix the hyperparameter-selection protocol.** The paper reports the best timestep after sweeping t = 0:300. This is potentially problematic unless the timestep is selected on a separate validation/calibration set. The authors should clearly separate validation and test data, report the exact selection procedure, and ideally provide performance curves over timesteps rather than only best-case values.
4. **Improve baseline fairness.**
The paper should justify why only norm-based variants of some baselines are evaluated, and should avoid broad superiority claims if stronger classifier-based or text-conditioned variants are omitted.
5. **Clarify held-out set construction for pre-trained models.**
For experiments involving pre-trained models, the authors should explain how they ensure that the held-out set was not seen during pre-training. This is especially important for web-scale datasets where near-duplicates or overlapping samples may exist. The paper should describe whether deduplication, similarity filtering, or dataset provenance checks were performed.
6. **Report runtime / wall-clock cost, not only query count.** Query count is useful, but different attacks may have different preprocessing, latent encoding, reconstruction, or memory costs. Since efficiency is a central claim, wall-clock comparisons would strengthen the paper.

Other Minor problems：

1. **Typos and inconsistency.** Fix presentation issues and minor inconsistencies, including “Appendix ??” references, typos such as “Evaulation,” inconsistent naming of SimA-MC / Sim-MC, and possible inconsistency between “eight datasets,” “11 datasets,” and “15 member–held-out pairs.”
2. **Organization.** Most main result tables appear around Page 8, while the detailed analysis is mainly placed on Pages 10–11. This separation makes readers repeatedly switch between tables and text.
3. **Improve writing precision**. For example, claims such as “without successful MIA, model inversion is impossible” should be softened, since training-data extraction or memorized-sample generation may occur without an explicit membership classifier.

---

> ### Author Response · Authors · 2026-06-07
> **Response for Review AMdZ**
>
> Thank you for your time and helpful comments. We provide responses here with respect to the points mentioned in "Requested Changes," though several are addressed in the general response.
>
> - **Preliminaries and Threat Model:** We will add a "Preliminaries and Threat Model" subsection in Section 3 of our revised manuscript, which will formally define the notation, the exact access required by the attacker (gray-box), arbitrary timestep queries, conditioning information, and the latent representation for the LDMs. We apologize for not clarifying these in the original manuscript.
>
> - **Baseline Fairness:** Although SimA without a norm can be adapted to classifier-based MIA, we focus on norm-based and unconditional variants. We acknowledge in the limitations that training auxiliary attack models (classifier-based MIAs) might yield higher success rates, but they may overfit to dataset-specific idiosyncrasies in a particular model instance, and require both a training set and potentially a higher computational cost. Admittedly, our own method requires validation data, but as the attack model expands, this training-data requirement also expands. We are not completely opposed to this, but we do feel there is high value in very low sample complexity / very low parameter attacks, both in theory and practice.
>
> - **Held-Out Set Construction for Pre-Trained Models:** In the experiments on the Stable Diffusion model, MSCOCO and Pokemon are not included in the training dataset (LAION-Aesthetics v2 5+). The selection criteria for LAION-2B-MultiTranslated are in Appendix C, including the aesthetic score, similarity score, and other criteria. We will restructure this section to enhance readability in our revised manuscript.
>
> - **Softening Claims:** We agree that our statement "without successful MIA, model inversion is impossible" was too strong. We have revised this to state that MIA often serves as a foundational step or strong indicator of vulnerability for downstream data extraction.
>
> - **Organization:** We have restructured the manuscript to bring the tables closer to their respective textual analyses.
>
> - **Runtime / Wall-Clock Cost:** Addressed in the general response.
>
> - **Hyperparameter-Selection Protocol:** Addressed in the general response.

---

### Review · Reviewer_1Wv8 · 2026-05-13

**Summary Of Contributions:**

The paper proposes a simple score-based membership inference attack for diffusion models. Instead of relying on reconstruction trajectories or multi-step denoising errors, the method uses the norm of the denoiser-predicted noise/score as the attack statistic. The key intuition is that, for overfitted diffusion models, the score norm around training samples tends to be smaller because the local denoising mean collapses toward the training point. The paper further introduces a Monte Carlo variant, SimA-MC, which averages the statistic over noisy perturbations for improved stability.

The main strengths are the simplicity of the attack, the clear connection between score estimation and membership leakage, and the broad empirical evaluation across DDPMs, LDMs, Guided Diffusion, and Stable Diffusion. The results show that SimA is competitive or stronger than prior diffusion MIA baselines while requiring fewer queries, and SimA-MC further improves low-FPR performance in most settings.

The main weaknesses are that some experimental choices need clearer justification, especially timestep selection, threshold calibration, and the fairness of reporting best swept timesteps across methods. The theoretical analysis is intuitive and useful, but still relies on assumptions about local density geometry and empirical overfitting that are not fully validated. Some results, such as the weaker SimA-MC performance on Guided Diffusion and the comparable rather than superior performance on Stable Diffusion pretraining, also deserve more discussion.

**Audience:**

Yes

**Audience Explanation:**

The findings are interesting and insightful. The claims are solid and supported by the experiments.

**Broader Impact Concerns:**

This work studies membership inference attacks against diffusion models and therefore has clear privacy implications. The paper frames the method as an auditing tool, which is appropriate, but the broader impact discussion should more explicitly address the dual-use nature of the attack. In particular, the authors should discuss how the released code and checkpoints could be used responsibly, what threat model the attack enables, and how model developers might use SimA to evaluate memorization risk before deployment.

I do not see broader impact concerns that would prevent publication, but the paper should include a clearer statement on responsible use, privacy auditing, and possible mitigation directions.

**Claims And Evidence:**

Yes

**Claims Explanation:**

Overall, the core claims are supported by the experiments. The paper evaluates SimA and SimA-MC against several diffusion-specific MIA baselines, including PIA, SecMI, PFAMI, and Loss, across multiple datasets and model families. The reported results generally support the claim that the score norm alone is a strong membership signal and that SimA provides a highly efficient alternative to reconstruction-based attacks.

However, the support is not equally strong for all claims. The claim of state-of-the-art performance is convincing for many DDPM/LDM and fine-tuned Stable Diffusion settings, especially when considering TPR@1%FPR for SimA-MC. Still, the results are more mixed for Guided Diffusion and the Stable Diffusion pretraining setting. The paper should therefore phrase the strongest claims more carefully, distinguishing single-query efficiency, average empirical strength, and universal SOTA performance.

The theoretical derivation is helpful and gives a plausible explanation for why score norms reveal membership. Yet the argument depends on local KDE-like behavior, finite-sample overfitting, and assumptions about on-manifold versus off-manifold points. These assumptions are reasonable but would benefit from stronger empirical validation, such as directly measuring score-norm distributions across timesteps, training stages, and degrees of memorization.

**Requested Changes:**

1. Clarify the timestep selection protocol. Since the paper sweeps timesteps and reports the best value for several methods, it should explain whether this tuning uses a validation set disjoint from the final test set. This is important for avoiding optimistic estimates.

2. Provide more details on threshold calibration. The attack uses a threshold for membership decisions, but the paper should clearly state what data are used to calibrate it and whether the same calibration budget is available to all baselines.

3. Temper the SOTA claim. SimA-MC is not uniformly best in every setting, and SimA is sometimes comparable rather than superior. The paper should distinguish "strong and efficient" from "universally state-of-the-art."

4. Add ablations on timestep sensitivity and norm choice. Since the paper uses the $ℓ_4$ norm for SimA and sweeps timesteps, it would be useful to show how performance changes across $ℓ_2$ , $ℓ_4$ , and different t values.

5. Discuss the Guided Diffusion exception more carefully. SimA-MC underperforms SimA and SecMI in some Guided Diffusion metrics, which weakens the general claim that Monte Carlo averaging consistently improves performance.

6. Although the paper focuses on MIAs against generative image models, the related work section would benefit from a broader discussion of recent membership inference studies on language models and multimodal language models. In particular, the following works are closely related and should at least be discussed:

[1] Detecting pretraining data from large language models (ICLR 2024)

[2] Membership inference attacks against large vision-language models (NeurIPS 2024)

[3] Vid-sme: membership inference attacks against large video understanding models (NeurIPS 2025)

[4] Membership inference attacks against vision-language models (Usenix Security 2025)

7. Fix minor writing issues and typos, such as "Evaulation" (page 10), "pretrianed" (page 15), and a few awkward phrases in the experimental section.

---

> ### Author Response · Authors · 2026-06-07
> **Response for Review 1Wv8**
>
> Thank you for your time and helpful comments. We provide responses here with respect to the points mentioned in "Requested Changes," though several are addressed in the general response.
>
> - **Threshold Calibration Details:** The threshold is chosen to maximize the ASR for each method on each dataset. We are re-running the experiments for timestep calibration, and we will use the same procedure to calibrate the threshold. Please see the general response for more experimental details.
>
> - **Ablations and Norm Choice:** We provided the comparison table for CIFAR-10 in the following, and will update the plots for more datasets in the revised manuscript.
> | Time | 0 | 20 | 40 | 60 | 80 | 100 | 120 | 140 | 160 | 180 | 200 | 220 | 240 | 260 | 280 | 300
> |---|---|---|---|---|---|---|---|---|---|---|---|---|---|---|---|---|
> | L4 norm | 46.78 | 51.11 | 63.41 | 73.91 | 80.67 | 84.75 | 87.28 | 88.89 | 89.83 | 90.32 | 90.45 | 90.16 | 89.46 | 88.25 | 86.51 | 84.18
> | L2 norm | 46.78 | 50.88 | 61.75 | 70.77 | 76.80 | 80.63 | 83.09 | 84.66 | 85.52 | 85.81 | 85.61 | 84.84 | 83.57 | 81.80 | 79.60 | 77.00
>
> - **Related Work on LLMs/VLMs:** Thank you for providing these excellent references. We will update our Related Work section to broaden the scope beyond image generation, discussing how membership inference behaves in LLMs and multimodal/video models, incorporating all four suggested citations.
>
> - **Typos:** We have gone through our manuscript again and tried to clear it of any remaining problems, fixing "Evaulation," "pretrianed," etc. in our revised manuscript. Our apologies for the cruft.
>
> - **Guided Diffusion Exception:** Addressed in general response, point 2.
>
> - **Clarification on Timestep Selection:** Addressed in the general response.
>
> - **Contradictions in Claims:** Addressed in the general response.

---

### Review · Reviewer_fzSD · 2026-05-29

**Summary Of Contributions:**

- This paper proposes SimA and SimA-MC, a single-query membership inference attack on diffusion models whose decision statistic is simply the norm of the predicted noise, $\lVert \hat{\epsilon}_\theta(x,t) \rVert$.
- Empirically show that SimA and SimA-MC provide top performance on standard datasets and independently trained base models.

**Audience:**

Yes

**Audience Explanation:**

The privacy of diffusion models is a major concern as other generative models and the findings of this paper about a simpler membership inference attack would be of interest to the targeted audience.

**Broader Impact Concerns:**

Needs to add a broader impact/ethics section since this is an attack paper.

**Claims And Evidence:**

No

**Claims Explanation:**

The overall methodology is ok but several claims are not supported by the evidence as presented:
- In Abstract,
  > Its Monte Carlo variant (SimA-MC) exhibits state-of-the-art performance across all experiments, significantly outperforming baseline methods in terms of TPR@1%FPR.

  In Table 3, SimA-MC performs worse than single-query SimA in terms of all metrics ASR/AUC/TPR, and specifically, both SimA and SimA-MC perform much worse compared to $\text{SecMI}_{\text{stat}}$ in terms of TPR@1%FPR.

- In Abstract,
  > SimA consistently achieves superior performance across variants of DMs and the Latent Diffusion Models (LDMs) on eight different
datasets.

  In Table 4, SimA (*) is not the SOTA under multiple settings. On Flickr30K, $\text{SecMI}_{\text{stat}}$ outperforms single-query SimA and SimA-MC(#mc=10) on all three metrics; only SimA-MC(#mc=30), which is the most query-expensive variant, outperforms.

- In general, the comparison between SimA and SimA-MC is not clear. Per the introduction,
  > To enhance stability, we further propose its Monte Carlo extension (SimA-MC). While involving a higher
computational cost due to additional model queries, SimA-MC further advances upon the current SoTA
across all experiments, notably boosting the TPR@1%FPR.

  - How much more computational cost for SimA-MC?
  - In Table 3, SimA-MC performs worse than single-query SimA in terms of all metrics ASR/AUC/TPR, and specifically, both SimA and SimA-MC perform much worse compared to $\text{SecMI}_{\text{stat}}$ in terms of TPR@1%FPR.
  - The robustness of benefits from additional queries have to be better adjusted.

- In section 4.2,
  > we swept t= 0:300 and reported the best-performing value for each method (PFAMI is
timestep-free and therefore needed no sweep).

  where selecting $t$ requires the groundtruth labels that the gray-box threat model does not grant. As a comparison, the paper calibrates the threshold $\tau$ on held-out data (Eq 12) but applies no such discipline to $t$.

**Requested Changes:**

Please refer to the previous section for details.

- Calibrate the claims to the tables: "SimA-MC … SoTA across all experiments" is contradicted by Table 3 (worse than SimA; both below SecMI on TPR), and "superior across LDMs" by Table 4.
- Quantify SimA-MC's cost (FLOPs/wall-clock, not just #Query; #mc=30 > SecMI's 12 queries) and state when the extra queries help vs hurt.
- Fix the timestep selection: the best $t$ reported from the sweep over $t \in [0,300]$ (Sec 4.2) is chosen using the evaluated set's ground-truth labels. Apply the same held-out methodology used for the threshold $\tau$ (Eq 12) and report at a fixed (or label-free) $t$, and show how performance varies with $t$.
- The overall writing needs to be improved a lot:
  - the broken cross-reference ("Appendix ??", Sec 4.1);
  - the dataset-count mismatch (abstract "eight" vs Sec 4.2 "11")
  - Appendix E.3, why does VAE/VQGAN appear here?
  - Table labels consistency: In Tables 1, 2, 3, the MC rows are labeled "SimA-MC (#mc=10)" / "SimA-MC (#mc=30)". In Table 4 the exact same rows are labeled "SimA (#mc=10)" / "SimA (#mc=30)", bringing possible confusion.

---

> ### Author Response · Authors · 2026-06-07
> **Response for Review fzSD**
>
> Thank you for your time and helpful comments. Regarding the requested changes, the first three points are all addressed in the general response (i.e., **Contradiction in Claims**, **Quantification of SimA-MC's Cost**, and **Time-step Selection**), and we will update our manuscript to reflect those changes.
>
> We also thank you for the writing and formatting fixes. We have gone through the manuscript carefully again and tried to identify any remaining issues for the updated version.

---

### Author Response · Authors · 2026-06-02
**General Response to All Reviewers for The Concerns Raised by Many Reviewers**

We sincerely thank all the reviewers for their constructive feedback and for recognizing the simplicity, theoretical intuition, and efficiency of our proposed single-query attack (SimA) and its Monte Carlo variant (SimA-MC).

Based on the reviewers' feedback, we have made several critical revisions to the manuscript. Below, we address the common concerns raised by multiple reviewers. **We will address point-by-point responses to individual comments under the individual response panel in the coming days**

## 1. Timestep Selection Protocol

**Concern (All Reviewers):** The protocol for sweeping and selecting the optimal timestep $t$ using the evaluated set's ground-truth labels risks information leakage and optimistic estimates.

**Response & Revision:**

We agree with this assessment. The optimal $t$ relies on both the noise schedule and the dataset, unfortunately. We clarify that we did not have a separate validation set to calibrate optimal $t$; **we have now rerun the experiments with a separate validation set for optimal $t$.** We would also like to additionally clarify that each method has a single $t$ value, this is *not* selected per datapoint, but selected once per [validation] dataset per method.

**Updated experimental procedure:** The original member and non-member sets were each randomly partitioned into two disjoint subsets of equal size, one used for validation and the other for testing. We will incorporate this clarification into the revised manuscript. We will also include the updated tables (on the new test set which excludes the validation set) in the manuscript. These experiments, especially for slower/Monte Carlo methods, will take additional time, but we expect them to complete before the end of the discussion period; we will add another post once the tables have been updated.

Preliminarily, the scores are nearly identical, as the selected optimal $t$ is similar or identical between validation and test sets (evaluated *post-hoc*), and the AUROC curves are very smooth with respect to $t$, at least at the bin granularity we tested. Plots of these curves will be added in the appendix of the revised manuscript.

## 2. Calibration of Claims (State-of-the-Art vs. Competitive)

**Concern (All Reviewers):** Claims such as "superior performance across all variants" and "state-of-the-art across all experiments" are contradicted by some results (e.g., Guided Diffusion, Table 3, and Stable Diffusion pretraining).

**Response & Revision:**
We broadly agree, and we have carefully re-calibrated the language throughout the abstract, introduction, and conclusion. We now frame SimA as a *highly efficient and competitive* alternative to reconstruction-based attacks that performs the best in many experiments, rather than universally superior. The original claim was written before the addition of the Guided Diffusion model results, we apologize for the statement's inaccuracy.

The text will now explicitly acknowledge settings where SimA-MC underperforms SecMI or where single-query SimA is preferable (e.g., Guided Diffusion). We expect SimA-MC to approach SimA as the number of Monte Carlo (MC) samples increases, and eventually to surpass it. In most cases, mc_samples = 10 is sufficient to match or exceed the performance of SimA. However, this behavior is strongly dependent on the data dimensionality. Very high-dimensional settings, such as Guided Diffusion ($3 \times 256 \times 256$), require a substantially larger number of MC samples, which is computationally prohibitive in our setting. This curse of dimensionality is alleviated when moving to Latent Diffusion Models (LDMs), where the effective dimensionality is significantly reduced; this is precisely why SimA-MC remains effective on Stable Diffusion, a variant of LDMs.

We have modified our claim to (effectively) "SimA-MC with 30 MC samples achieves state-of-the-art performance in **most** experiments, particularly in terms of TPR@1%FPR. Meanwhile, SimA serves as an effective point-estimate surrogate that attains highly competitive performance with only a single query." We will put these modifications in our revised manuscript.

---

> ### Author Response · Authors · 2026-06-02
> **General Response to All Reviewers for The Concerns Raised by Many Reviewers (Cont.)**
>
> ## 3. Quantification of Computational Cost
>
> **Concern (Reviewers fzSD & AMdZ):** The computational cost of SimA-MC needs to be quantified beyond just query count (#mc), specifically including wall-clock time and FLOPs.
>
> **Response & Revision:** We provide the table here. It will also be updated in the revised manuscript.
>
> **Table: Computational efficiency of MIA methods on CIFAR-10.** UNet NFE denotes the number of UNet function evaluations per query image. GFLOPs are computed as UNet FLOPs per forward pass (measured via `fvcore`) multiplied by NFE. Although there might be other FLOPs besides the forward pass, the UNet call dominates the FLOPs. Wall-clock time is measured on a single GPU averaged over 50,000 samples.
>
> | Method | UNet NFE ↓ | GFLOPs ↓ | Latency (ms/sample) ↓ |
> |--------|-----------:|---------:|----------------------:|
> | SimA   | 1  | 6.24   | 0.71  |
> | Loss   | 1  | 6.24   | 0.71  |
> | PIA    | 2  | 12.48  | 1.41  |
> | SecMI  | 12 | 74.87  | 8.45  |
> | PFAMI  | 20 | 124.78 | 14.08 |
>
> The results show SimA and Loss are the two most efficient attack statistics, which is consistent with the claim in the manuscript.
>
> ## 4. Broader Impact and Ethics Statement
>
> **Concern (Reviewers fzSD & 1Wv8):** As an attack paper, a Broader Impact statement addressing the dual-use nature and privacy implications is necessary.
>
> **Response & Revision:** We have added a comprehensive "Broader Impacts and Ethics" section at the end of the manuscript. We explicitly discuss the dual-use nature of SimA: while it serves as a lightweight privacy auditing tool for model developers to quantify memorization, it could theoretically be deployed by malicious actors. We outline responsible disclosure practices and potential mitigation strategies for developers.
>
> **Again, we will upload the revised manuscipt once the experiments are finished by the end of the discussion period. We will address point-by-point responses to individual comments under the individual response panel in the coming days**

---

> ### Author Response · Authors · 2026-06-13
> **Manuscript Updated**
>
> We have updated the manuscript. Additions are highlighted in blue text, and deletions are marked with a red strikethrough. We have rerun all experiments, calibrating $t$ and $\tau$ on a separate validation set. As mentioned above, the results are slightly changed. Please review the revised version.

---

### Decision · Action_Editor_UH9a · 2026-07-04

**Recommendation:** Accept as is

**Audience:**

Yes

**Audience Explanation:**

Membership inference attack for diffusion models is an important and interesting topic.

**Claims And Evidence:**

Yes

**Claims Explanation:**

The paper proposes a simple and elegant score-based membership inference attack for diffusion models, using the norm of the denoiser/score output as an efficient attack statistic. The theoretical motivation is clear, and the empirical evaluation is reasonably broad, covering DDPMs, LDMs, Guided Diffusion, and Stable Diffusion settings, where the proposed method is generally competitive or better than prior diffusion MIA baselines. Overall, the work is sound, useful, and relevant to the wide audience.

Although one reviewer tend to reject the paper due to the limited novelty. However, based on the evaluation criteria, novelty should not be a criteria for evaluation. So my final decision is accept.